# Isothiocyanate-Rich Extracts from Cauliflower (*Brassica oleracea* Var. Botrytis) and Radish (*Raphanus sativus*) Inhibited Metabolic Activity and Induced ROS in Selected Human HCT116 and HT-29 Colorectal Cancer Cells

**DOI:** 10.3390/ijerph192214919

**Published:** 2022-11-13

**Authors:** Mardey Liceth Cuellar-Nuñez, Ivan Luzardo-Ocampo, Sarah Lee-Martínez, Michelle Larrauri-Rodríguez, Guadalupe Zaldívar-Lelo de Larrea, Rosa Martha Pérez-Serrano, Nicolás Camacho-Calderón

**Affiliations:** 1Advanced Biomedical Research Center, School of Medicine, Universidad Autónoma de Querétaro, Queretaro 76140, Mexico; 2Instituto de Neurobiología, Universidad Nacional Autónoma de México (UNAM), Queretaro 76230, Mexico; 3Licenciatura en Medicina General, Facultad de Medicina, Universidad Autónoma de Querétaro, Queretaro 76176, Mexico

**Keywords:** cauliflower (*Brassica oleracea*), colon cancer, in silico, isothiocyanates, LDH, MTT, radish (*Raphanus sativus*)

## Abstract

Cruciferous vegetables such as cauliflower and radish contain isothiocyanates exhibiting chemoprotective effects in vitro and in vivo. This research aimed to assess the impact of cauliflower (CIE) and radish (RIE) isothiocyanate extracts on the metabolic activity, intracellular reactive oxygen species (ROS), and LDH production of selected human colorectal adenocarcinoma cells (HCT116 and HT-29 for early and late colon cancer development, respectively). Non-cancerous colon cells (CCD-33Co) were used as a cytotoxicity control. The CIE samples displayed the highest allyl isothiocyanate (AITC: 12.55 µg/g) contents, whereas RIE was the most abundant in benzyl isothiocyanate (BITC: 15.35 µg/g). Both extracts effectively inhibited HCT116 and HT-29 metabolic activity, but the CIE impact was higher than that of RIE on HCT116 (IC_50_: 0.56 mg/mL). Assays using the half-inhibitory concentrations (IC_50_) of all treatments, including AITC and BITC, displayed increased (*p* < 0.05) LDH (absorbance: 0.25–0.40 nm) and ROS release (1190–1697 relative fluorescence units) in both cell lines. BITC showed the highest in silico binding affinity with all the tested colorectal cancer molecular markers (NF-kB, β-catenin, and NRF2-NFE2). The theoretical evaluation of AITC and BITC bioavailability showed high values for both compounds. The results indicate that CIE and RIE extracts display chemopreventive effects in vitro, but additional experiments are needed to validate their effects.

## 1. Introduction

Colorectal cancer (CRC) is one of the most frequent worldwide diseases in terms of morbidity and mortality rates. In 2020, 0.94 million deaths occurred, and 1.93 million new CRC cases were diagnosed. An increase of 3.9 million new CRC cases is predicted for 2040 [1]. At least 80% of cases are generated sporadically, while the remanent cases are linked to family history [2].

One of the most important strategies for treating many cancer types is chemotherapy. However, the high concentration of chemically synthesized anticancer drugs has been shown to present several side effects, including nausea, hair loss, tiredness, loss of appetite, shortness of breath, diarrhea, skin allergies, and infertility [3]. The search for an alternative that could reduce these effects is a target in chemoprevention.

Dietary intake of fruits and vegetables is one of the most important chemopreventive strategies. Notably, it has been reported that consuming cruciferous vegetables exerts health benefits. Cruciferous plants and their natural derivatives have the potential to inhibit and reduce the risk of cancer [4]. Among cruciferous vegetables, the *Brassicaceae* family is one of the most well known because it contains some of the most consumed vegetables globally and provides significant health benefits [5]. Some of the most well-known *Brassicaceae* are cauliflower (*Brassica oleracea* var. botrytis), broccoli (*B. oleracea* var. italica), Brussels sprouts (*B. oleracea* var. gemmifera), bok choy (*B. rapa* sbsp. chinensis), cabbage (*B. oleracea* var. capitata), turnip greens (*B. rapa* var. rapa), and radishes (*Raphanus sativus*). Other well-known cruciferous are moringa (*Moringa oleifera*) and other dark green leafy vegetables.

Cruciferous vegetables are rich in nutrients and bioactive compounds, including flavonoids and phenolics acids; carotenoids such as β-carotene, zeaxanthin, and lutein; vitamins A, B_6_, C, E, and K; folate; minerals; and dietary fiber [6]. Additionally, cruciferous plants contain chemical sulfur compounds known as glucosinolates and S-methylcysteine sulfoxide, which are responsible for the bitter taste and pungent aroma of cruciferous plants. During the physiological transformation of glucosinolate-rich foods, biologically active compounds such as isothiocyanates, among other components, are formed [7]. In aqueous environments, glucosinolates are hydrolyzed by the enzyme β-thioglucoside-glucohydrolase (myrosinase), leading to the formation of isothiocyanates [8].

Dietary isothiocyanates (ITCs) are biochemical compounds that can modify certain pathways at the cellular level and inhibit cancer progression by modulating the epigenome [9]. Among the described ITC types, some of the most important are benzyl isothiocyanate (BITC), allyl isothiocyanate (AITC), phenyl isothiocyanate (PITC), and sulforaphane (SFN) [10]. BITC and AITC are naturally present in vegetables such as watercress (*Nasturtium officinale*), cabbage, cauliflower, radish, kale (*B. oleracea* var. sabellica), and broccoli [10]. Particularly for radish, 4-methylthio-3-butenyl isothiocyanate is considered the main ITC, but other compounds, such as AITC, BITC, and phenetyl isothiocyanate, have also been linked to its pungent flavor and biological properties [11]. Regarding cauliflower, its odor and typical flavor are associated with volatile compounds, including ITCs, such as AITC, 2-methylpropyl isothiocyanate, but-3-enyl isothiocyanate, 3-methylbutyl isothiocyanate, and 4-methylpentyl isothiocyanate [12]. Reports have confirmed the functionality of these compounds in interfering in stages of the inflammatory process and inhibiting or delaying oncogenesis by altering different molecular processes. Moreover, ITCs inhibit cancer formation in various organs in vivo, including the bladder, mammary, colon, liver, and lung [13,14,15]. Isothiocyanates can inhibit the activation of some complexes, such as NF-κB, which is involved in inflammation processes and responses to stress [16]. Others are related to the synthesis of the NRF2 protein through the activation of the transcription factor NFE2L2, which has antioxidant effects [17]. In addition, they have epigenetic mechanisms of action, such as histone deacetylation and the inhibition of DNA methylation and microRNA expression [9,18].

Studies on BITC and AITC have shown that these compounds activate NRF2, a protein that controls the expression and induction of cytoprotective genes and induces the accumulation of autophagic molecules such as LC3BII and p62 in AGS cells [19]. In addition, it has been reported that BITC induced apoptotic cell death by inhibiting AKT in breast cancer cells.

Mechanisms delaying the progression or advancement of colorectal cancer, such as cell death by apoptosis, the activation of the antioxidant system, and cell cycle arrest, are of particular interest in the search for natural alternatives for the treatment of colon cancer. As there are no reports of the impact of cauliflower or radish isothiocyanate extracts on colorectal cancer, this research aimed to evaluate their effects on the metabolic activity, intracellular reactive oxygen species (ROS) release, and LDH production of two representative human colorectal adenocarcinoma cells (HCT116 and HT-29 for early and late colon cancer development, respectively).

## 2. Materials and Methods

### 2.1. Plant Material

Edible cauliflower (*B. oleracea* var. botrytis) heads and radish (*R. sativus*) roots were acquired from local markets of Queretaro (Mexico). The parts were selected, washed (200 ppm sodium hypochlorite), and cut, and the tissues were freeze-dried in a Stoppering Tray Dryer (7755013, Labconco, Kansas City, MO, USA) at −40 °C and 133 Bar for seven days. Subsequently, the freeze-dried material was ground and stored in sealed bags at −80 °C until further analyses were carried out.

### 2.2. Glucosinolate Extraction and Enzymatic Conversion into Isothiocyanates

The extraction of glucosinolates was conducted as reported by Förster et al. [20]. Briefly, the freeze-dried material (20 mg cauliflower heads or radish roots) was mixed with 750 µL of 70 % *v*/*v* methanol, and the mixture was placed in 1.5 mL microcentrifuge tubes with locking snap caps and heated at 80 °C for 10 min in a Thermomixer (Eppendorf, Hamburg, Germany). The extraction was repeated twice, using 500 µL each time.

After cooling (22 ± 1 °C), the samples were centrifuged (16,000× *g*), and supernatants were concentrated in a vacuum concentrator (Savant SpeedVac DNA130, ThermoScientific, Waltham, MA, USA) to a 150 µL volume. Then, 200 µL of 0.4 M barium acetate was added, and the volume was increased to 1 mL using MilliQ water. The samples were incubated for 30 min at room temperature (22 ± 1 °C) and filtered through centrifugation using 0.22 µm Costar Spin X tubes (CLS8160, Sigma-Aldrich, Sant Louis, MO, USA) in a centrifuge (16,000× *g*). Then, one milliliter of the filtrate was incubated with 0.05 U myrosinase (thioglucosidase from *Sinapis alba*, Sigma-Aldrich) for 8 h at 37 °C to allow enzymatic conversion from the original glucosinolates (sinigrin and glucotropaeolin) to their derived isothiocyanates (AITC and BITC, respectively). The resulting hydrolyzed samples (glucosinolates extracts) were filtered again by centrifugation (16,000× *g*) using 0.22 µm Costar Spin X tubes, and the filtrates were stored at −80 °C until further analysis. Extracts from cauliflower and radish were named cauliflower isothiocyanate (CIE) and radish isothiocyanate (RIE) extracts, respectively.

### 2.3. Identification and Quantification of Standard Isothiocyanates in CIE and RIE Extracts

The identification and quantification of standard isothiocyanates (allyl isothiocyanate, AITC; benzyl isothiocyanate, BITC) were carried out by adapting a previously reported method [21]. Briefly, a Prominence high-performance liquid chromatography (HPLC) system (Shimadzu, Columbia, MD, USA) using two LC 20AD pumps, an IL-20A autoinjector, a DGU-20As degasser, an SPD-20AUV-VIS detector, and a CBM-20A communication BUS module was used, and compounds were separated in a C18 Inertsil reverse-phase column (250 mm, 4.6 mm, 05 µm granule size) (GL Sciences, Torrance, CA, USA). Two mobile phases were used: 100% methanol (A) and an aqueous solution of 0.005 M tetrabutylammonium bisulfate (TBAB). The sample (15 µL) was separated using the following conditions: 12% A for 2 min, 35% A for 20 min, 50% A for 20 min, and 100% A for 10 min. A standard curve of HPLC-grade BITC and AITC standards was prepared. Isothiocyanates were expressed as µg equivalents of each isothiocyanate (AITC or BITC)/g sample. Three different RIE and CIE extracts were prepared, and replicates of each extract, using two injections, were quantified (*n* = 18). The validation of the conducted HPLC method is shown in Appendix A. Validation was carried out as recommended by the International Council for Harmonization (ICH) [22]. Briefly, linearity was assessed using a working range of each standard (0–100 µg/mL), and each concentration was injected three times to obtain the area under the curve (AUC). Data were used to plot a linear regression curve (y = mx + b, where “y” is the response or AUC, “x” is the assayed concentration of AITC or BITC, “m” is the slope of the curve, and “b” is the y-intercept). The limit of detection (LOD) and limit of quantification (LOQ) were considered the limits at which AITC or BITC can be reliably detected or quantified using a 3:1 signal-to-noise ratio, considering the standard deviation of the calibration curve (σ) and the slope of the calibration curve (S): LOD = 3.3σ/S and LOQ = 10σ/S. To determine the recovery, a minimum of 6 determinations of 100% of the test concentrations (low concentrations: 15.20 and 14.90 µg/mL; medium concentrations: 35.71 and 40.23 µg/mL; and high concentrations: 60.22 and 62.14 µg/mL) were prepared, spiked into the processed sample as indicated in Section 2.2, and injected in the HPLC system. The AUC for each test was used to calculate the AITC or BITC concentration using the previously calculated standard curves. The percentage recovery and the relative standard deviation (% RSD) were also calculated. For each assay, separate weights of the standards were evaluated. Together with the suggested procedures of ICH, the usage of AITC and BITC standards was based on previous reports using these pure standards spiked into the sample for recovery assays [23,24].

### 2.4. Cell Culture

Human colorectal adenocarcinoma HT-29 (ATCC HTB-38), colorectal carcinoma HCT116 (ATCC CCL-247), and large intestine epithelial (CCD-33Co ATCC CRL-1539) cells were acquired from American Type Culture Collection (ATCC). The cells were maintained under proper conditions (humidified 37 °C atmosphere at 5% CO_2_) and minimum essential medium (MEM) (Gibco™, Waltham, MA, USA) supplemented with 10 % fetal bovine serum (Gibco), 1% antibiotic–antimycotic (100X, Gibco), and 1 % sodium pyruvate (100X, Gibco). CCD-33Co cells were used as a cytotoxicity control for the metabolic activity assay.

#### 2.4.1. Assessment of the Impact of Isothiocyanate Extracts and Isothiocyanate Standards on the Metabolic Activity of Cells

The 3-(4,5-dimethylthiazol-2-yl)-5-(3-carboxymethoxyphenyl)-2-(4-sulphophenyl)-2H-tetrazolium (MTS) assay was used to assess the metabolic activity of the cells exposed to CIE, RIE, AITC, and BITC. Briefly, the cells (HTC116, HT29, and CCD-33Co; 1 × 10^4^ cells/well, 96-well plates) were seeded for 24 h. The medium was then replaced with CIE/RIE (0.125, 0.250, 0.500, and 1.000 mg/mL) or AITC/BITC (10, 25, 50, 100, 150, and 200 µM) treatments, dissolved in MEM + 10% FBS, for 24 h. Then, the CellTiter 96^®^ AQ_ueous_ One Solution Cell Proliferation Assay (MTS) kit (Promega Corporation, Madison, WI, USA) was used following the manufacturer’s conditions. The half-inhibitory concentration (IC_50_) of each extract was calculated following a three-parameter dose–response equation for biological assays provided by GraphPad Prism v. 8.2.1. software (Dotmatics, Stortford, UK). The three-parameter dose–response equation was used, considering that the data were perfectly adjusted to this model (Appendix A). Moreover, the selected non-linear model reflects IC_50_ calculations based on the fact that real-life studies do not always behave linearly [25]. In addition, the model has been successfully used in colorectal cancer cell lines to find the IC_50_ of selected treatments [26,27]. The results were also expressed as the inhibition percentage of the negative control (untreated cells), and oxaliplatin-treated cells were used as the positive control since this substance has been validated as an effective cytotoxic drug in colon cancer treatment [28]. The oxaliplatin IC_50_ was also calculated for HT29 and HCT116 cells.

#### 2.4.2. Impact of the Isothiocyanate Extracts and Standards on Lactate Dehydrogenase (LDH) Activity

The cells (HT29 and HCT116, 1 × 10^4^ cells/well) were seeded in 96-well plates for 24 h, and the medium was then replaced with the IC_50_ concentrations of CIE, RIE, BITC, and AITC treatments for 24 h. After the treatments, the medium was collected in sterile tubes and centrifuged (5000× *g*, 22 °C), and supernatants (50 µL) were added to a new 96-well plate and mixed with 50 µL of the LDH reaction mix (Pierce LDH Cytotoxicity Assay Kit, ThermoScientific) for 30 min at 22 °C. After the incubation, 50 µL of the provided stop solution was added, and the fluorescence was measured in a spectrophotometer (Varioskan, ThermoScientific) at 490 nm (680 nm was used to subtract the background signal from the instrument).

#### 2.4.3. Effect of Isothiocyanate Extracts’ and Standards’ IC_50_ on Intracellular Reactive Oxygen Species (ROS) Production

ROS production in the IC_50_-treatment-challenged cells was quantified using the 2′,7′-dichlorodihydrofluorescein acetate reagent (H_2_DCFDA, Invitrogen, Waltham, MA, USA). Briefly, the cells (HT29 and HCT116, 1 × 10^4^ cells/well) were seeded in black 96-well plates for 24 h. After the incubation, the cells were treated with the IC_50_ concentrations of CIE, RIE, BITC, and AITC for 24 h. Then, 5 µM H_2_DCFDA was added, and the cells were incubated for 45 min in the dark. The fluorescence was measured at 480/530 nm excitation/emission intensities in a Varioskan multiplate reader (ThermoScientific). The results were expressed as relative fluorescence units (RFU).

### 2.5. In Silico Analysis of BITC and AITC Inhibition Affinity with Colon-Cancer-Associated Protein Markers

To evaluate the potential inhibition (inhibition binding affinity) with reported colorectal cancer proteins markers targeted by ligands such as BITC (PubChem CID: 2346) and AITC (PubChem CID: 5971), the 3D protein structures of the p50/p65 NF-κB dimer (PDB ID: 1NFI), β-catenin (PDB ID: 2GL7), and NRF2-NFE2 (PDB ID: 3WN7) were obtained [29]. To simulate inhibitions, either each molecule was downloaded with an inhibitor (such as NFκB docked with IκBα [30]; NRF2 with the Kelch-like EACH-associated protein or KEAP-1 [31]), or the molecule was previously docked with a proposed inhibitor (such as β-catenin against a potential developed inhibitor) [32], and AITC/BITC were docked in the same position of the inhibitor. The ligands and the molecular targets were prepared by eliminating water molecules and additional chemical compounds in Discovery Studio Visualizer v. 19.1.0.18287 software (Dassault Systèmes, Vélizy-Villacoublay, France). Then, hydrogens and Gasteiger charges were assigned using AutoDock Vina [33], and the interaction was modeled following the suggested procedure of Luna-Vital, Weiss, and Gonzalez-De Mejia [34]. The visualization of the most probable poses showing the lowest binding affinity was carried out in Discovery Studio Visualizer (Dassault-Systèmes). The theoretical IC_50_ values [IC_50 (t)_] were calculated using the Gibbs free energy equation reported elsewhere: ∆G: -RT·lnK, where ∆G is Gibbs free energy in kcal/mol, R is the Raoult constant (1.986 kcal/mol·K), and T is the ideal reaction temperature (298 K). A scheme of the conducted in silico procedures is presented in Appendix A.

### 2.6. In Silico Evaluation of BITC and AITC Intestinal Absorption and Bioavailability Parameters

An in silico assessment of predictive BITC and AITC intestinal absorption, as this is a critical parameter to guarantee their potential bioactivity [35], was conducted as previously reported [36]. For this, an evaluation of the binding affinity between the isothiocyanates (AITC and BITC) and a model of the intestinal membrane were used. A 1:1:1 lipid membrane made from 1-palmitoyl-2-oleosyl-sn-glycero-3-phosphocholine (POPC), 1-palmitoyl-2-oleosyl-sn-glycero-3-phosphoetanoamine, and cholesterol was generated in the Membrane Builder utility from PlayMolecule (https://playmolecule.org, accessed on 18 September 2022) [37], with a 50 Å size on the “x”- and “y”-axes. The resulting PDB file was prepared at pH 7.4 in the Protein Prepare utility from PlayMolecule software. Docking exercises were conducted as previously indicated in Section 3.5. The predictive permeability was screened for the blood–brain barrier (BBB), the Caco-2 cell model (Caco-2 M), human intestinal absorption (HIA), human oral bioavailability (HOB), and bioavailability using the simplified molecular input line entry specifications (SMILES) of AITC and BITC generated in PubChem (https://pubchem.ncbi.nlm.nih.gov/, accessed on 18 September 2022) and computed in admetSAR 2.0 online software (http://lmmd.ecust.edu.cn/admetsar2, accessed on 18 September 2022). The intuitive and multiple absorptions of compounds were plotted in a “Boiled Egg” diagram plotting the atomistic interpretation of the fragmental system of Wildman and Crippen for lipophilicity (WLOGP) and the topological polar surface area for apparent polarity (TPSA) in the SwissADME software (http://www.swissadme.ch/index.php#4, accessed on 18 September 2022) [38] (Appendix A).

### 2.7. Statistical Analysis

The results are expressed as the mean ± SD of three independent experiments in triplicate. An Analysis of Variance (ANOVA) was performed, followed by a post hoc Tukey–Kramer test, establishing the significance level at *p* < 0.05. The analyses were carried out in JMP v. 16 (SAS, Cary, NC, USA).

## 3. Results

### 3.1. Quantification of BITC and AITC Isothiocyanates in CIE and RIE

The results presented in Figure 1 show the representative chromatograms of the cauliflower (CIE) (Figure 1A) and radish (RIE) (Figure 1B) isothiocyanate extracts. Both chromatograms displayed several peaks, where allyl isothiocyanate (AITC) and benzyl isothiocyanate (BITC) were identified based on the retention times presented in Figure 1C. Standard curves calculated for AITC and BITC allowed the quantification of each isothiocyanate (Figure 1D), where no differences (*p* < 0.05) were observed for BITC, but CIE displayed the highest AITC amount.

### 3.2. Impact of Treatments on the Cell Metabolic Activity

Figure 2 displays the impact of the treatments (CIE and RIE) on the metabolic activity of HCT116 (Figure 2A), HT-29 (Figure 2B), and CCD-33Co (Figure 2C) cells. As observed, CIE showed the highest inhibition (*p* < 0.05) in HCT116 at all assayed concentrations, no differences were shown for either treatment in HT-29 cells, and a low inhibition of metabolic activity was observed in CCD-33Co, indicating that the extracts were not cytotoxic to the latter cells. The calculation of the IC_50_ values for both cancer cell lines (Figure 2D) indicated that CIE exhibited the strongest inhibitory effect on both cell lines (+23.22% and +7.58% for HCT116 and HT-29 cells, respectively). Moreover, AITC and BITC were also assayed as representative standards, with AITC being the most effective in HCT116 and BITC being the most effective in HT-29. Oxaliplatin (OX) was evaluated as a representative drug treatment, showing the lowest IC_50_ concentrations.

### 3.3. Effect of Isothiocyanate Extracts on LDH Activity in Cells

The effect of the IC_50_ concentrations of CIE, RIE, AITC, and BITC on the LDH activity of HCT116 and HT-29 cells is presented in Figure 3. Overall, the same LDH-inducing trend was present for both cell lines, but the treatments induced higher LDH values in HCT116 than in HT-29. There were no differences between CIE, AITC, and BITC, but RIE showed the lowest effect.

### 3.4. Assessment of ROS Release in Cells Treated with the Extracts

Both CIE and BITC stimulated the highest intracellular ROS production in both cell lines (Figure 4), but no differences were shown for the HT-29 cell line between the mentioned treatments and AITC (*p* > 0.05). Intracellular ROS production was higher for all treatments compared to untreated cells, and the treatments were more effective in inducing ROS in the HCT116 cell line.

### 3.5. In Silico Potential Inhibition Evaluation of AITC and BITC, and Assessment of Their Bioavailability and Intestinal Absorption

Based on their Gibbs free energy calculations, a lower binding affinity was observed between AITC and molecular cancer targets (Figure 5) compared to BITC (Figure 6), considering that ΔG values for BITC are lower than those of AITC. No differences in the energy values were present for AITC, whereas BITC showed the highest affinity with NF-κB, followed by β-catenin and NRF2-NFE2. In both compounds (AITC and BITC), the van der Waals bonds were the most common. In addition, the calculation of the theoretical IC_50_ values in the interaction with the selected molecular cancers indicated similar values. However, a higher BITC amount is potentially required to induce a response with NF-κB (Figure 6).

Theoretical evaluation of the compounds’ affinity with a complex model of the intestinal membrane (Figure 7) showed that BITC has a stronger affinity for the membrane lipids than AITC (based on its lowest ΔG values), suggesting more interactions between this molecule and the membrane.

As shown by the predicted bioavailability values (Figure 8), AITC is more suitable for oral bioavailability, as all of the evaluated parameters fit the ideal conditions for human oral bioavailability, highlighted in the colored zone (Figure 8A). In contrast, BITC did not achieve the ideal unsaturation values. Despite differences between the two compounds for potential absorption in the Caco-2 model (Caco-2 M), human intestinal absorption (HIA), and human oral bioavailability (HOB), where AITC displayed the highest values, both compounds received the same bioavailability score (BS: 0.55) (Figure 8B).

The additional prediction of drug-like targeting indicated a major ability of BITC to display more active sites for the recognition of macrophage migration inhibition factor (MIF), transient receptor potential cation channel subfamily A member 1 (TRPA1), and indoleamine-2,3-dioxygenase (IDO) compared to AITC (Figure 8C). The “Boiled Egg” diagram for predicting the gastrointestinal absorption and blood–brain barrier (BBB) crossing indicated a similar behavior between the two compounds, successfully crossing the intestinal epithelium and BBB.

## 4. Discussion

Isothiocyanates are a class of natural and almost exclusively plant-derived compounds biologically stored as glucosinolates. However, the co-existing plant myrosinase (thioglucosidase glycohydrolase) catalyzes their conversion into isothiocyanates [39]. Nonetheless, some of these ingested compounds can reach the colon and be processed by microbial β-thioglucosidase, as human homologs of myrosinase have not yet been described [40]. Cruciferous vegetables such as cauliflower (*Brassica oleracea* var. botrytis) and radish (*Raphanus sativus*) are the most common sources of glucosinolates and the main dietary source of isothiocyanates [41]. Broccoli is the most well-known source of glucosinolate-producing isothiocyanates, exhibiting many biological effects in several conditions, such as pancreatic cancer, colon cancer, leukemia, and prostate cancer; additional health benefits in osteoporosis and osteoarthritis; and anti-inflammatory properties [42]. However, additional isothiocyanates derived from diverse plant species still require in vitro and in vivo assays to fully understand and elucidate their health benefits since most of their properties remain unknown [43].

Allyl isothiocyanate (AITC) has been defined as one of the main isothiocyanates produced from sinigrin, the predominant glucosinolate in cruciferous vegetables. It has been reported that high temperatures can destroy myrosinase, the natural glucosinolate-degrading enzyme, and sinigrin is particularly resistant to high-temperature extraction procedures [44]. Human exposure to AITC is widespread and abundant, and its biological properties, such as anticancer effects, are desirable, as AITC is highly bioavailable [45]. On the other hand, BITC is a common isothiocyanate from the *Brassica* and *Raphanus* genera and exhibits important chemopreventive effects against several types of cancer, particularly breast, lung, and liver cancer [46]. Like sinigrin, glucotropaeolin, the original BITC source, is also resistant to high-temperature extraction methods [44]. The calculated AITC values for CIE extracts are similar to those previously reported for raw cauliflower (10.29 µg/g), and the values are more than 3-fold higher than AITC contents in broccoli (3.18 µg/g) [47]. Moreover, AITC is the major sinigrin-derived isothiocyanate in the total volatile fraction of cauliflower, representing 2–4% of the total fraction [48]. AITC has been identified as one of the most significant isothiocyanates in *Raphanus,* together with 4-methylthio-3-butenyl isothiocyanate [49]. AITC contents have been reported as present in *Raphanus sativus* var. niger (0.0025 % of AITC) and butyl isothiocyanate-containing essential oil. In another report, black radish was reported to have 14.5–23.1 µg AITC/mL for each 2–8 g powder/100 mL (not specified) [50], which is close to the values indicated in this research. The evaluation of root, leaves, and stem acetone and hexane extracts of *R. sativus* yielded 0.013–0.21 mg/g AITC and 0.001–0.023 mg/g BITC [51]. More recently, Liu et al. [5] indicated that cauliflower contained some of the highest AITC levels among 18 *Brassicaceae* plants, whereas radish displayed some of the lowest levels for the same isothiocyanate. Moreover, high recovery rates (87.4–96.30%) were achieved when AITC was enzymatically released from sinigrin [52], validating the enzymatic procedure conducted in this research. On the other hand, BITC content has been scarcely reported in vegetables. However, it is one of the predominant isothiocyanates in cauliflower and radish (4.5 and 2.2%, respectively) among the total myrosinase-mediated glucosinolate hydrolysis products [53].

As isothiocyanates have demonstrated anticarcinogenic effects on several cell lines, the impact of cauliflower (CIE) and radish (RIE) isothiocyanate extracts was evaluated in selected colon cancer cell lines. In contrast, a non-cancerous epithelial colon cell line (CCD-33Co) was used as a cytotoxicity control. The inhibition of the metabolic activity of these cells agreed with reports suggesting the inhibition of cell viability and proliferation due to isothiocyanates. The colorectal cancer cell lines used in this study were selected based on their genotypic characteristics exhibiting differences in colorectal cancer development. Hence, HCT116 cells represent an early stage of colon cancer, having defects in mismatch repair proteins (MMR) and mutations in mutL homolog 1 protein MLH1, which is essential in DNA repair [54]. In contrast, HT-29 cells exemplify an advanced colon cancer stage, as the expression of MMR proteins is normal [55]. A report evaluating several Cruciferae plants on HCT116 cells showed IC_50_ > 50 µg/mL for broccoli (*B. oleracea* var. italica) flowers and Chinese radish (*R. sativus* var. longipinnatus) roots [56]. Since CIE and RIE did not present IC_50_ < 20 µg/mL, they could not be considered active extracts, according to the National Cancer Research Institute [57]. However, the inhibition of metabolic activity that the extracts showed might be related to their AITC or BITC contents since these compounds have been linked to strong antiproliferative effects in vitro. For instance, AITC (5 and 10 µM) showed suppressive effects on the cell invasion abilities of epithelial growth factor (EGF)-induced HT-29 cells [58]. Since CIE contained a higher AITC abundance, it could be inferred that these extracts possess a major anti-metastatic ability, although more in vitro and in vivo studies are needed to confirm these effects. The pure benzyl isothiocyanate standard at doses >0.25 µM (3.7 × 10^−5^ mg/mL) reduced HT29 cell viability to <80%. Although the authors did not provide the IC_50_ value, this one was close to 2.5 µM (3.7 × 10^−4^ mg/mL), a value that is much lower than the dose reported in this research. On a different colorectal cancer cell line (Caco-2), Chinese kale (*Brassica oleracea* var. alboglabra L.) displayed cytotoxic effects >20 % for 150–250 µg/mL doses, and the effect was attributed to the total isothiocyanate content of the seedlings (1.06 mmol/100 g dry weight) [59]. In HT-29 cells, AITC and BITC displayed IC_50_ of 5.4 µM (53.5 × 10^−3^ mg/mL) and 10 µM (1.49 × 10^−3^ mg/mL), respectively [60].

As lactate dehydrogenase is only released by necrotic cells, it has been validated as an indicator of this cell death mechanism. A glucosinolate-rich hydrolyzed extract from *Moringa oleifera* only showed significant effects on LDH release for HT-29 but not for HCT116 cells, compared to their untreated counterparts [16]. Other authors evaluating the impact of *Moringa oleifera* also found increased LDH release in HCT116, Caco-2, and HCT116 p53^−/−^ cells, and the activity was presumably linked to the isothiocyanate content of the leaves [61]. Since most isothiocyanates are released from vegetal sources due to bacterial enzymatic activity, it could be inferred that the colonic microbiota activity could release these compounds to exert their necrotic effect.

One of the best-known effects of isothiocyanates is the suppression of tumor growth by generating intracellular ROS or inducing cell cycle arrest to induce apoptosis [62]. Moreover, isothiocyanates modulate phase I and II enzymes to reduce the bioactivation of carcinogens, leading to less binding of carcinogens to DNA and fewer pro-mutagenic effects [63]. Hence, both the extracts and standards could induce ROS, although CIE and BITC effects were more substantial on HCT116, which could be related to the major BITC contents in CIE. There are no reports on the ROS-inducing effects of CIE and RIE or isothiocyanate extracts on colorectal cancer cell lines. Nevertheless, the challenge of human metastatic bone PC3 cells with cauliflower seed extracts showed an increased ROS effect [64]. Sulforaphane from broccoli, belonging to a class of glucosinolates, can trigger apoptosis in several cancer cell lines and can increase ROS levels to induce the cytosolic release of cytochrome C to further trigger apoptosis [65]. The generation of ROS in breast cancer cells (MDA-MB-231, MCF-7, and MCF-10A) due to BITC treatment (25 µM) was reported to be partially responsible for pro-apoptotic effects linked to BITC [66]. Moreover, BITC treatments on HT-29 similarly released ROS to oxaliplatin (OX), whereas BITC-induced ROS activity in HCT116 cells was lower than OX but higher than in the untreated cells [16].

Expanding on the potential molecular mechanisms of isothiocyanates from the extracts (AITC and BITC) linked to colorectal cancer, the in silico analysis agreed with the reported effects of BITC on the induction of IκB kinase phosphorylation, IκB-a, and p65, allowing NF-kB translocation into the nucleus [67]. Experimentally, it was found that AITC inhibited lipopolysaccharide (LPS) NF-kB activation and induced IκBα phosphorylation in HT-29 cells [68]. The in silico interaction between AITC and NRF2 was explored by Rajakumar et al. [69], and the authors suggested NRF2 induction by AITC in 7,12-dimethylbenz(a) anthracene (DMBA)-induced mammary tumors in vivo. Similarly, an interaction between AITC and NF-κB was also reported [70], but the authors did not explain the effects from the inhibition perspective. BITC inhibited the DNA binding of NF-κB in HT-29 cells and reduced the phosphorylation of associated upstream proteins such as c-Jun terminal kinase 1 and 2 (JNK1/2), extracellular signal-regulated kinases 1 and 2 (ERK 1/2), phosphatidylinositol 3-kinase (PI3K), and protein kinase C (PKC) [46]. In addition, BITC is associated with decreased β-catenin-dependent cyclin D1 transcription and activates the PI3K/AKT survival pathway by inhibiting tyrosine phosphatase 1B, helping in the pro-apoptotic mechanisms initiated by NF-κB [71]. Autophagy mechanisms have also been linked to BITC, with the up-regulation of NRF2 and NF-E2 [72], which could agree with the reported pro-autophagy mechanisms of *Moringa oleifera*-derived colonic metabolites in HT-29 cells [73]. Hence, BITC could interact at several molecular levels with the evaluated in silico targets.

To ensure that AITC and BITC can reach the colon or target organs, a high bioavailability must be reached, and both compounds showed high oral and intestinal bioavailability scores, indicating the high ability of these compounds to either be absorbed in the small intestine if orally administered or be highly bioaccessible after their microbial release from dietary glucosinolates. AITC is one of the most studied glucosinolates, and more than 90% oral bioavailability has been reported [74]. It has been reported that BITC bioavailability is affected by variabilities in resorption mechanisms and the excretion of BITC metabolites. However, BITC displays an overall rapid and high bioavailability since its urinary metabolites rapidly disappear after 6 h [75]. Further research evaluating the bioaccessibility and bioavailability of the targeted isothiocyanates is encouraged, aiming to evaluate their natural presence under gastrointestinal conditions, potential release, and biological effects. Hence, it has been reported that based on bioinformatic predictions, BITC and AITC, together with phenetyl-isothiocyanate, phenyl-isothiocyanate, 4-isothiocyanabut-1-ene, 7-isothiocyanatohept-1-ene, iberin, and ethyl isothiocyanate, are isothiocyanates with the largest biological spectrum and are promising for pharmacological research [76].

Surprisingly, the potential interaction of AITC and BITC with MIF, TRPA1, and IDO could reveal a new research line exploring the effect of several AITC and BITC doses on these molecular targets. For instance, MIF regulates the progression from colitis to colorectal cancer through its involvement in cell differentiation and *il-17* gene regulation and participates in tumor growth and development in vivo [77]. Modifications of BITC at the N-terminal catalytic proline residue alter its MIF modulation and increase its inhibitory activity against MIF [78]. TRP channels are currently one of the main focuses in colorectal cancer, as their dysregulation influences the proliferation, migration, invasion, and pro-apoptotic ability of cancer cells, together with the advantage of being expressed extracellularly, so compounds do not need to enter the cells to exert their effects [79].

## 5. Conclusions

The results obtained in this research suggested that cauliflower and radish isothiocyanate extracts effectively inhibited HT-29 and HCT116 metabolic activity (>80%), increased the production of intracellular ROS (41.63–93.29%, compared to untreated cells), and induced LDH release (+21.02 to 77.23%, compared to untreated cells). Although it has been reported that AITC and BITC have high bioavailability and the biological activities of CIE and RIE extracts could be linked to these compounds, further in vivo experiments must be conducted to prove their properties. The in silico modeling of individual isothiocyanates with molecular cancer targets may suggest the potential inhibition of NF-κB, β-catenin, and NRF2 proteins, but additional in vitro and in vivo experiments are needed to validate these effects. Moreover, since the evaluated assays are not exclusive to colorectal cancer cells, additional research involving other cancer cell types is needed.

## Figures and Tables

**Figure 1 ijerph-19-14919-f001:**
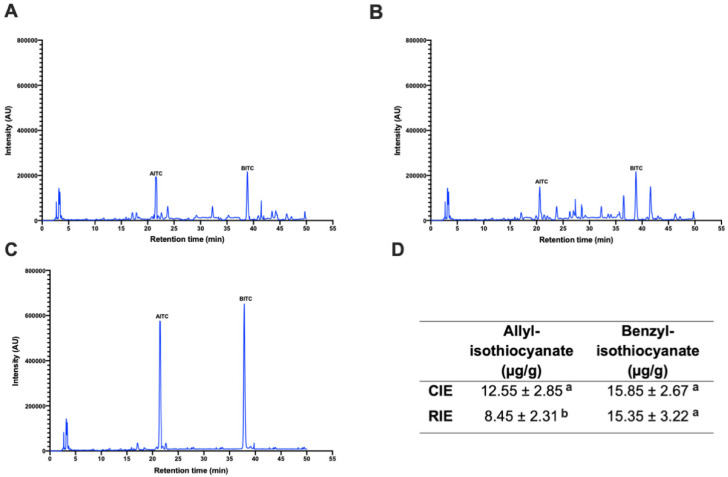
Identification and quantification of allyl isothiocyanate (AITC) and benzyl isothiocyanate (BITC) in the cauliflower isothiocyanate (CIE) and radish isothiocyanate (RIE) extracts. (**A**) Representative chromatogram from CIE; (**B**) representative chromatogram from RIE; (**C**) AITC and BITC standards’ representative chromatogram; (**D**) quantification of AITC and BITC contents in the CIE and RIE extracts. AU: Absorbance units. The results in (**D**) are the mean ± SD of three independent experiments in triplicate. Different letters in (**D**) indicate significant differences (*p* < 0.05) by the Tukey–Kramer test.

**Figure 2 ijerph-19-14919-f002:**
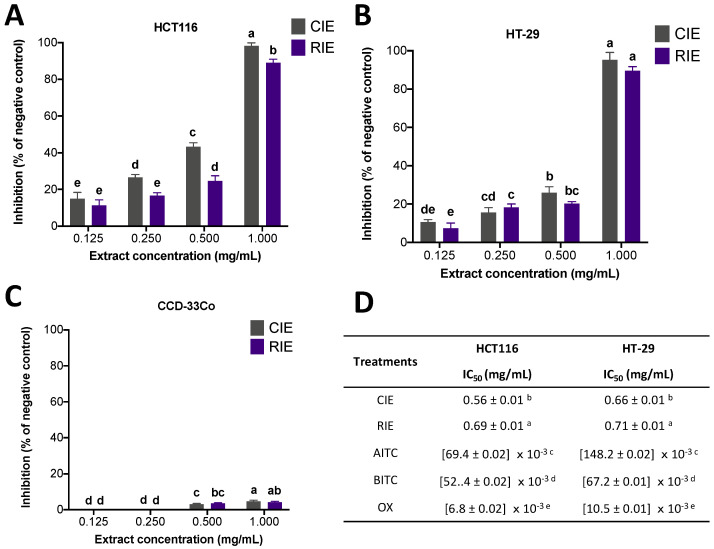
Evaluation of metabolic activity (MTS) in cells treated with CIE, RIE, AITC, BITC, and oxaliplatin (OX) treatments. Effect of CIE and RIE treatments on the metabolic activity inhibition of (**A**) HCT116, (**B**) HT-29, and (**C**) CCD-33Co cells. (**D**) Calculation of half-inhibitory concentrations (IC_50_, mg/mL) of the treatments on HCT116 and HT-29 cells. The results are expressed as the mean ± SD of three independent experiments in triplicate. Different letters indicate significant differences (*p* < 0.05) by the Tukey–Kramer test for each concentration (**A**–**C**) or between treatments (**D**).

**Figure 3 ijerph-19-14919-f003:**
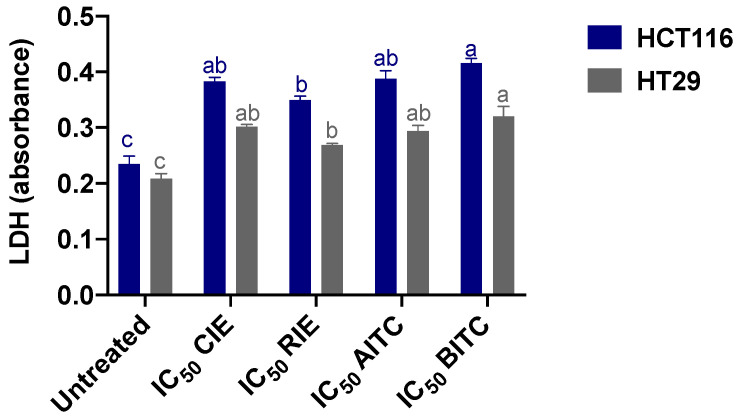
Lactate dehydrogenase (LDH) activity in HCT116 and HT29 treated with the half-inhibitory concentrations of cauliflower isothiocyanate (CIE) and radish isothiocyanate (RIE) extracts, allyl isothiocyanate (AITC), and benzyl isothiocyanate (BITC). The results are the mean ± SD of three independent experiments in triplicate. Different letters express significant differences (*p* < 0.05) by the Tukey–Kramer test for each cell line between treatments. Untreated cells corresponded to MEM-only-treated cells.

**Figure 4 ijerph-19-14919-f004:**
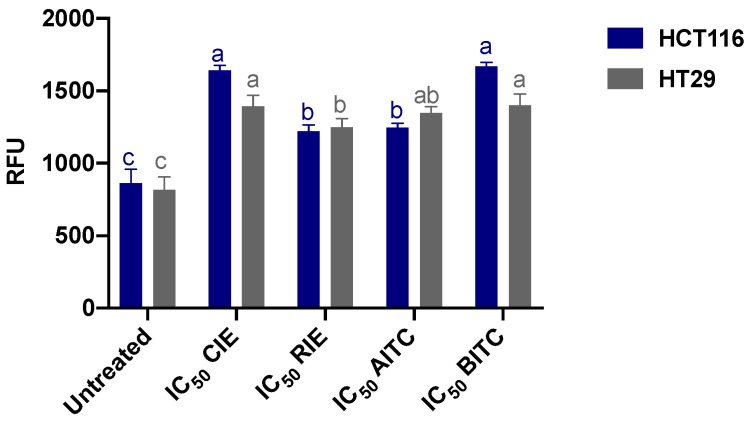
Intracellular reactive oxygen species (ROS) produced by HCT116 and HT29 cells after treatment with the half-inhibitory concentrations (IC_50_) of cauliflower and radish isothiocyanate extracts (CIE and RIE), allyl isothiocyanate (AITC), and benzyl isothiocyanate (BITC). The results are the mean ± SD of three independent experiments in triplicate. Different letters express significant differences (*p* < 0.05) by the Tukey–Kramer test for each cell line between treatments. Untreated cells corresponded to MEM-only-treated cells. RFU: Relative fluorescence units.

**Figure 5 ijerph-19-14919-f005:**
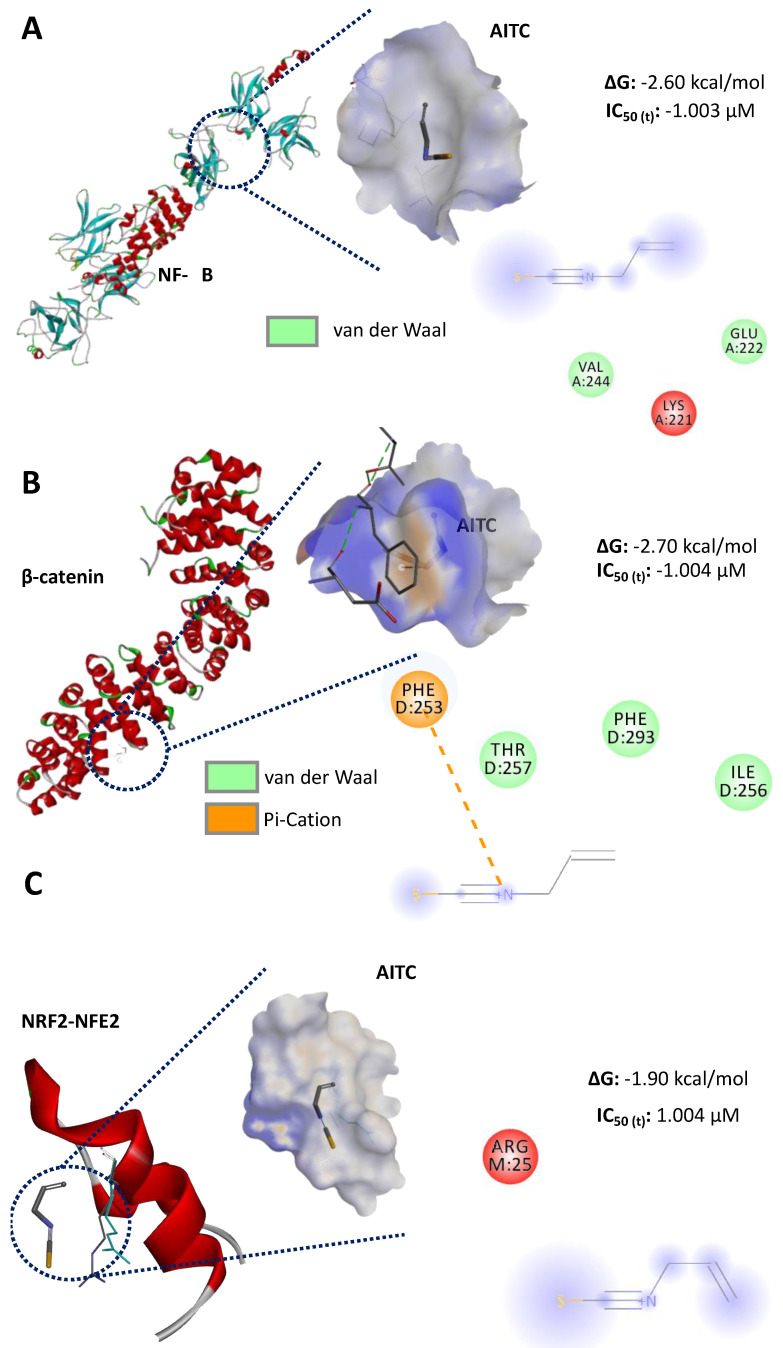
In silico inhibition evaluation of the binding affinity between allyl isothiocyanate (AITC) and molecular targets of colorectal cancer. (**A**) Binding between AITC and nuclear factor κB (NF-κB); (**B**) binding between AITC and β-catenin; (**C**) binding between AITC and NRF2-NFE2. ∆G: Gibbs free energy; IC_50 (t)_: theoretical half-inhibitory concentration (calculated using ∆G).

**Figure 6 ijerph-19-14919-f006:**
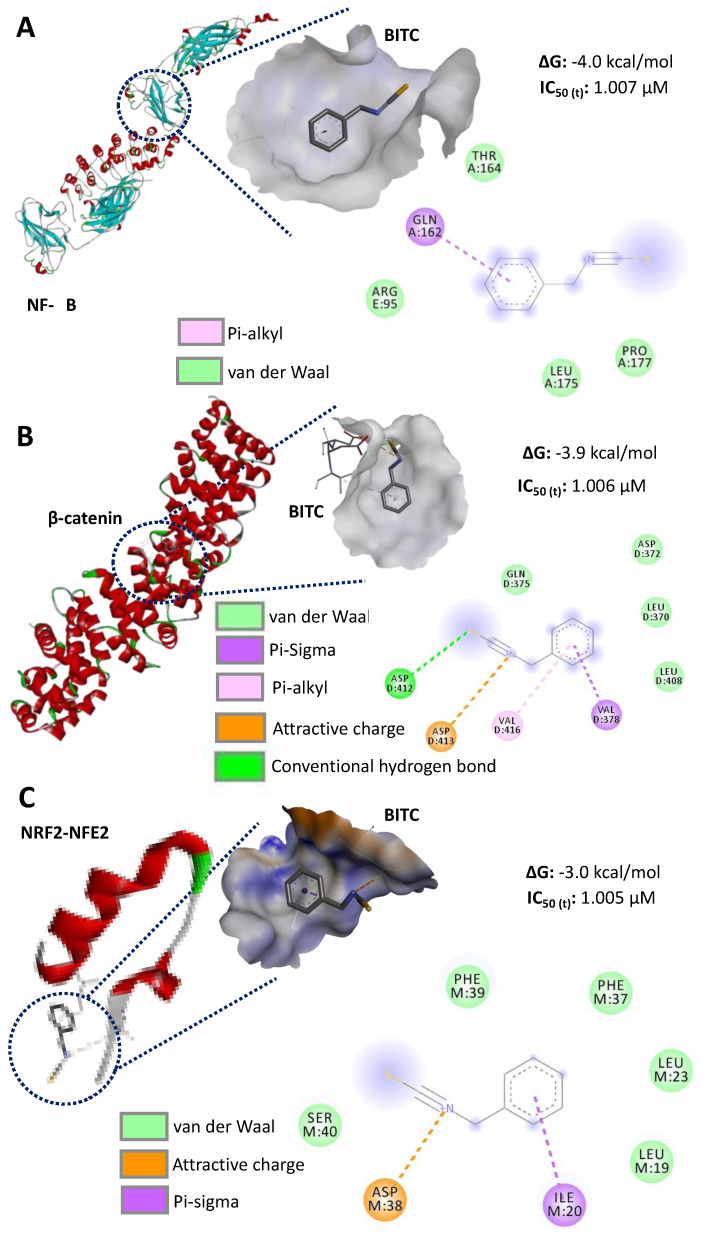
In silico inhibition evaluation of the binding affinity between benzyl isothiocyanate (BITC) and molecular targets of colorectal cancer cells. (**A**) Binding between BITC and nuclear factor κB (NF-κB); (**B**) binding between BITC and β-catenin; (**C**) binding between BITC and NRF2-NFE2. ΔG: Gibbs free energy; IC_50 (t)_: theoretical half-inhibitory concentration (calculated using ∆G).

**Figure 7 ijerph-19-14919-f007:**
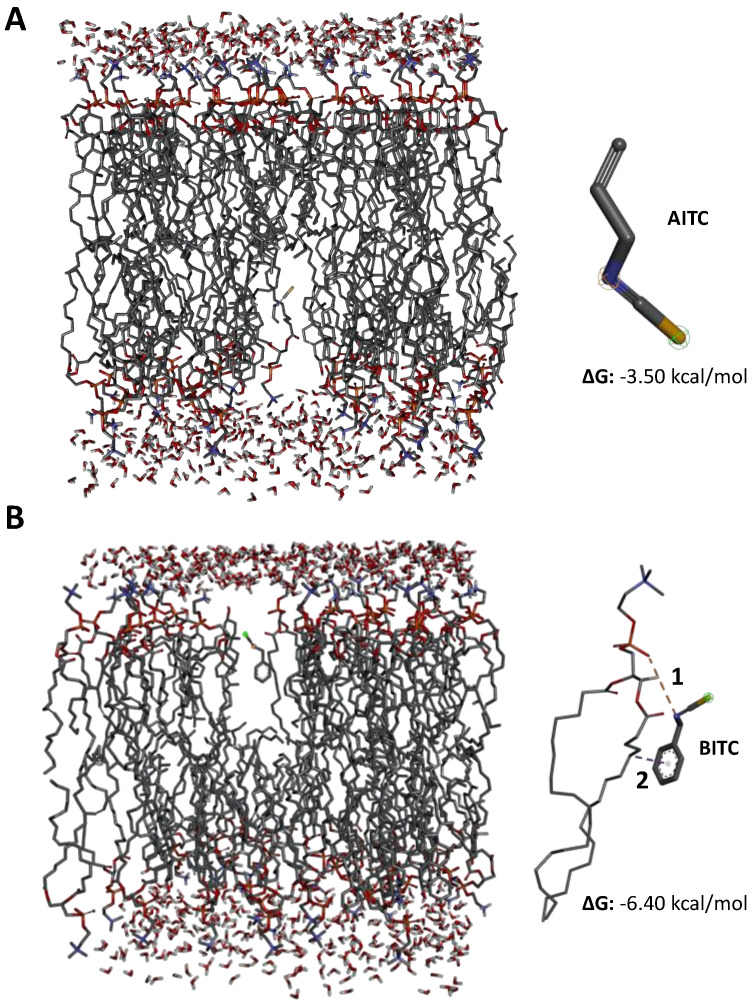
In silico screening of the binding affinity between a complex 1-palmitoyl-2-oleosyl-sn-glycero-3-phosphocholine (POPC), 1-palmitoyl-2-oleosyl-sn-glycero-3-phosphoetanoamine, and cholesterol membrane and (**A**) allyl isothiocyanate (AITC) and (**B**) benzyl-isothiocyanate (BITC). ΔG: Gibbs free energy; 1: attractive charge; 2: Pi-alkyl.

**Figure 8 ijerph-19-14919-f008:**
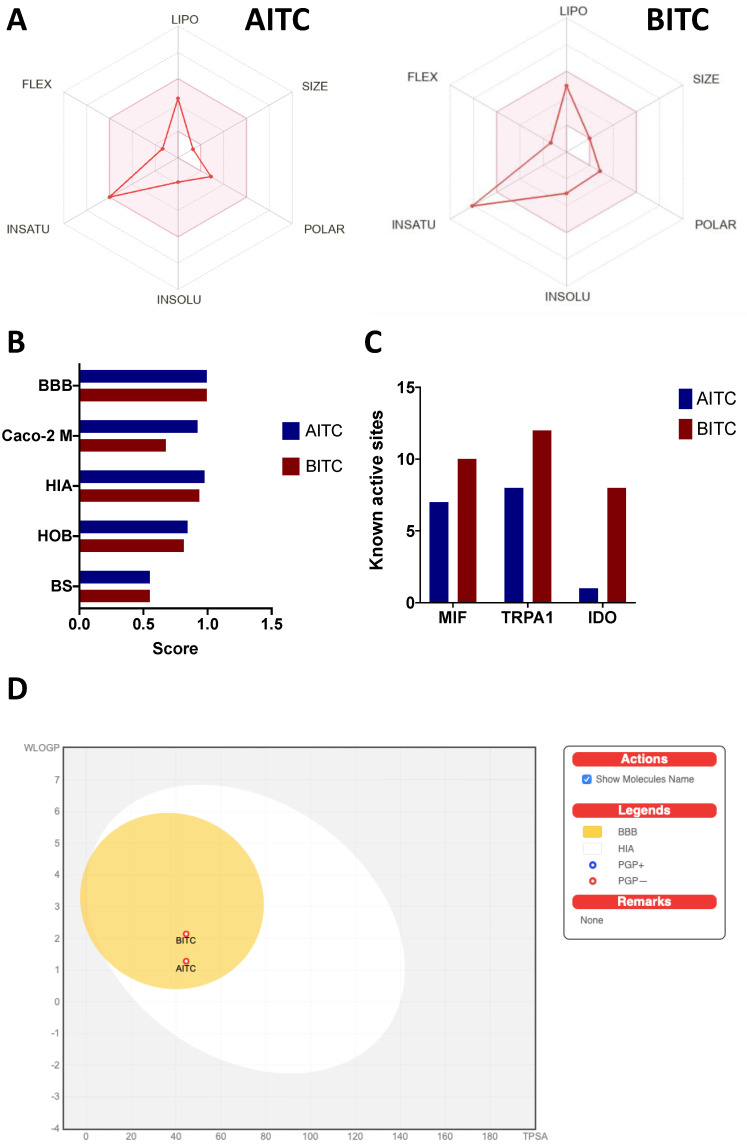
In silico prediction of bioavailability scores of AITC and BITC. Overall bioavailability plots of (**A**) AITC and BITC; (**B**) calculated scores of blood–brain barrier (BBB) permeability, Caco-2 cell model (Caco-2 M) permeability, human intestinal absorption (HIA), human oral bioavailability (HOB), and bioavailability score (BS) of AITC and BITC; (**C**) prediction of the best drug-targeting abilities of AITC and BITC based on their known active sites for macrophage migration inhibitor factor (MIF), transient receptor potential cation channel subfamily A member 1 (TRPA1), and indoleamine-2,3-dioxygenase (IDO); and (**D**) “Boiled Egg” model of AITC and BITC absorption plotting the atomistic interpretation of the fragmental system of Wildman and Crippen for lipophilicity (WLOGP) and the topological polar surface area for apparent polarity (TPSA). The colored zone in Figure 7A shows the suitable physicochemical space for HOB (lipophilicity: −0.7 to 5; size: 150–500 g/mol; polarity: 20–130 Å^2^; insolubility: −6 to 0; unsaturation: 0.25–1; flexibility: 0–9 rotatable bonds). FLEX: flexibility; INSATU: unsaturation; INSOLU: insolubility; LIPO: lipophilicity; POLAR: polarity; PGP: P-glycoprotein.

## Data Availability

Not applicable.

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
