# Peer review of "Isothiocyanate-Rich Extracts from Cauliflower (Brassica oleracea Var. Botrytis) and Radish (Raphanus sativus) Inhibited Metabolic Activity and Induced ROS in Selected Human HCT116 and HT-29 Colorectal Cancer Cells"

_ijerph, 2022, doi:10.3390/ijerph192214919_

Round 1
Reviewer 1 Report
Here is my comments;
Good efforts done by the authors in this research study. I would suggest for minor correction for this paper to be publish with the revision of comments given as follow:
- Suggest to change keyword to simple form and don’t repeat the same keyword as mentioned in the title.
- Refer to extraction method of the glucosinolate, the freeze-dried material (20 mg) was mixed with 750 μL of 70 % v/v methanol at 103 80 ºC and heated at 80 ºC for 10 min in a Thermomixer (Eppendorf, Hamburg, Germany).
The boiling point of methanol is about 78 °C. How you ensure the methanol extract is not vaporated and you can yield the methanol abstract?
- If the isothiocyanates is you target compound, how you explain high heating will destroy the myrosinase enzyme within the brassicae vegetables, and resulting yielding lower ITCs?
Please do have a read on https://plantmethods.biomedcentral.com/articles/10.1186/s13007-017-0164-8
- What is the three-parameter dose-response equation you mentioned in line 149. Please elaborate further and why is it being used to get IC50 result?
- Refer to line 195-196, The binding affinity between the selected compounds and a model of the intestinal membrane was used. This statement is confusing, suggest to add on one figure of flow chart about the whole process involved in in-silico analysis and evaluation.
- Why the presentative of IC50 in Figure 2 is present in log(concentration)? What is the actual concentration of CIE and RIE loaded in MTS analysis?
- The legend presented in Figure 5 and 6 are confusing. The type of bonding presented is read by the ---- line or the circle shape? both green colour legends are make reader to difficult to understand the contents.
- How do you determine the BITC showed stronger affinity from figure 7b compared with figure 7a?? Please explain further about this.
- There are two figure 7 in the manuscript. Please revise the last Figure.
- Suggest to shift the contents on line 306-312 after figure 7.
- Please double check on the correct figure number being used, revise line 321.
Author Response
Authors’ responses to Reviewer 1
- Reviewer: Good efforts done by the authors in this research study. I would suggest for minor correction for this paper to be publish with the revision of comments given as follow
- Authors’ response: We appreciate the reviewer’s comments.
- Reviewer: Suggest to change keyword to simple form and don’t repeat the same keyword as mentioned in the title.
- Authors’ response: Thank you for your comments. We have simplified the keywords as you kindly suggested. However, we decided to keep the raw materials since we believe they are crucial in this manuscript. Please refer to the revised manuscript.
Revised manuscript:
Page 1, Lines 32-33: Cauliflower (Brassica oleracea), colon cancer, in silico, isothiocyanates, LDH, MTT, radish (Raphanus sativus).
- Reviewer: Refer to extraction method of the glucosinolate, the freeze-dried material (20 mg) was mixed with 750 μL of 70 % v/v methanol at 80 ºC and heated at 80 ºC for 10 min in a Thermomixer (Eppendorf, Hamburg, Germany).
The boiling point of methanol is about 78 °C. How you ensure the methanol extract is not evaporated and you can yield the methanol abstract?
- Authors’ response: For the extraction, samples were placed in 1.8 mL tubes with a locking snap cap, ensuring the volatile methanol could not escape from the tubes. Originally, methanol was at room temperature (22 ± 1 ºC), and we apologize to the reviewer for this, as this was incorrectly indicated in the manuscript. After extraction, tubes were brought to room temperature (22 ± 1 ºC) and completely locked, so that no methanol could escaped. We have added this to the method. Please refer to the revised manuscript.
Revised manuscript:
Page 3, Lines 109-114: The extraction of glucosinolates was conducted, as reported by Förster et al. [20]. Briefly, the freeze-dried material (20 mg) was mixed with 750 µL of 70 % v/v methanol, the mixture was placed in 1.5 mL microcentrifuge tubes with a locking snap cap and heated at 80 ºC for 10 min in a Thermomixer (Eppendorf, Hamburg, Germany). The extraction was repeated twice using 500 µL each time. After cooling (22 ± 1 ºC), the samples were centrifuged (16000´ g)
- Reviewer: If the isothiocyanates is you target compound, how you explain high heating will destroy the myrosinase enzyme within the Brassicae vegetables, and resulting yielding lower ITCs? Please do have a read on https://plantmethods.biomedcentral.com/articles/10.1186/s13007-017-0164-8
- Authors’ response: Thank you for your comments and the provided reference. We are aware that boiling vegetables could destroy myrosinase enzyme, yielding a lower amount of ITCs, which are our target compounds. However, particularly for the assayed vegetable species in this research and the extraction method, boiling water or methanol has not significantly decreased sinigrin or glucotropaeolin (Doheny-Adams et al., 2017), the main glucosinolates producing AITC and BITC, respectively. Moreover, still boiling methanol is considered an effective method for glucosinolates extraction (Förster et al., 2015). We have added this information to the revised manuscript.
Revised manuscript
- Page 13, Lines 384-387: It has been reported that high temperatures can destroy myrosinase, the natural glucosinolates-degrading enzyme, and sinigrin is particularly resistant to high-temperature extraction procedures [41].
- Page 13, Lines 391-392: Like sinigrin, glucotropaeolin, the original BITC source, is also resistant to high-temperature extraction methods [41].
- Reviewer: What is the three-parameter dose-response equation you mentioned in line 149. Please elaborate further and why is it being used to get IC50 result?
- Authors’ response: The three-parameter dose-response equation is an automatic provided equation for biological samples in a dose-response situation from GraphPad Prism software. This model is a non-linear model that has been successfully used to find IC50 values in ex vivo models (Estes et al., 2007; Kendrick et al., 2008), pharmacogenomics studies integrating in vitro with genome analysis (Pozdeyev et al., 2016), and we have used this model testing green-synthesized gold nanoparticles against human HT29 and SW480 colon adenocarcinoma cell lines, obtaining IC50 values that matched with an increased caspase-3 response (Soto et al., 2022). Shen et al. (2020) indicated that the three-parameter dose-response equation was the most successful model when defining IC50 in colorectal cancer cells. Moreover, compared to linear models, the three parameters dose-response model provides a basis for understanding IC50 calculations reflecting that real-life studies are not well-behaved, e.g., linear responses (Lyles et al., 2008). We have added part of this information in the revised manuscript. In addition, our data almost perfectly fits within the model. We have added the curves in Supplementary Figure 2.
Revised manuscript
- Page 4, Lines 160-165: The three-parameter dose-response equation was used considering that data was perfectly adjusted to this model (Supplementary Figure S1). Moreover, the selected non-linear model reflects IC50 calculations based on the fact that real-life studies are not always linearly behaved [22]. In addition, the model has been successfully used in colorectal cancer cell lines to find IC50 of selected treatments [23,24].
References
Estes, J. M., Oliver, P. G., Straughn, J. M., Zhou, T., Wang, W., Grizzle, W. E., Alvarez, R. D., Stockard, C. R., LoBuglio, A. F., & Buchsbaum, D. J. (2007). Efficacy of anti-death receptor 5 (DR5) antibody (TRA-8) against primary human ovarian carcinoma using a novel ex vivo tissue slice model. Gynecologic Oncology, 105(2), 291–298. https://doi.org/10.1016/j.ygyno.2006.12.033
Kendrick, J. E., Straughn, J. M., Oliver, P. G., Wang, W., Nan, L., Grizzle, W. E., Stockard, C. R., Alvarez, R. D., & Buchsbaum, D. J. (2008). Anti-tumor activity of the TRA-8 anti-DR5 antibody in combination with cisplatin in an ex vivo human cervical cancer model. Gynecologic Oncology, 108(3), 591–597. https://doi.org/10.1016/j.ygyno.2007.11.039
Lyles, R. H., Poindexter, C., Evans, A., Brown, M., & Cooper, C. R. (2008). Nonlinear model-based estimates of IC50 for studies involving continuous therapeutic dose–response data. Contemporary Clinical Trials, 29(6), 878–886. https://doi.org/10.1016/j.cct.2008.05.009
Pozdeyev, N., Yoo, M., Mackie, R., Schweppe, R. E., Tan, A. C., & Haugen, B. R. (2016). Integrating heterogeneous drug sensitivity data from cancer pharmacogenomic studies. Oncotarget, 7(32), 51619–51625. https://doi.org/10.18632/oncotarget.10010
Shen, J., Li, L., Yang, T., Cohen, P. S., & Sun, G. (2020). Biphasic Mathematical Model of Cell–Drug Interaction That Separates Target-Specific and Off-Target Inhibition and Suggests Potent Targeted Drug Combinations for Multi-Driver Colorectal Cancer Cells. Cancers, 12(2), 436. https://doi.org/10.3390/cancers12020436
Soto, K. M., Luzardo-Ocampo, I., López-Romero, J. M., Mendoza, S., Loarca-Piña, G., Rivera-Muñoz, E. M., & Manzano-Ramírez, A. (2022). Gold Nanoparticles Synthesized with Common Mullein (Verbascum thapsus) and Castor Bean (Ricinus communis) Ethanolic Extracts Displayed Antiproliferative Effects and Induced Caspase 3 Activity in Human HT29 and SW480 Cancer Cells. Pharmaceutics, 14(10), 2069. https://doi.org/10.3390/pharmaceutics14102069
- Reviewer: Refer to line 195-196, The binding affinity between the selected compounds and a model of the intestinal membrane was used. This statement is confusing, suggest to add on one figure of flow chart about the whole process involved in in-silico analysis and evaluation.
- Authors’ response: We have re-phrased the statement and included a supplementary figure indicating an overall scheme of the in silico procedures. Please refer to the revised manuscript.
Revised manuscript:
- Page 5, Lines 213-214: A scheme of the conducted in silico procedures is presented in Supplementary Figure 2A.
- Page 5, Lines 219-220: For this, an evaluation of the binding affinity between the isothiocyanates (AITC and BITC) and a model of the intestinal membrane were used.
- Page 5, Line 235: (Supplementary Figure 1B and 1C).
- Reviewer: Why the presentative of IC50 in Figure 2 is present in log (concentration)? What is the actual concentration of CIE and RIE loaded in MTS analysis?
- Authors’ response: We have changed Fig. 2 for graphics using concentrations instead of Log(Concentration). Moreover, the actual CIE and RIE doses have been presented in Revised Figure 2.
Within the text, the CIE and RIE doses were presented on Page 4, Lines 154-155. Please refer to the revised manuscript to find Revised Figure 2.
- Reviewer: The legend presented in Figure 5 and 6 are confusing. The type of bonding presented is read by the ---- line or the circle shape? both green colour legends are make reader to difficult to understand the contents.
- Authors’ response: We apologize to the reviewer for the legends, and we have rephrased it in the Revised Manuscript. Regarding the bonding types, they refer to the dashed line, which we have indicated it in the Revised manuscript. For the color legends, unfortunately, the colors are the default software colors, and we cannot change them; however, the only figure having two different green colors now is Figure 6B, where the clearest one refers to van der Waals (for the neighboring atoms). The darkest one refers to Conventional hydrogen-bond as a dashed line.
Revised manuscript
- Page 9, Lines 315-318: In silico inhibition evaluation of the binding affinity between allyl isothiocyanate (AITC) and molecular targets of colorectal cancer. (A) Binding between AITC and nuclear factor kB (NF-kB); (B) Binding between AITC and b-catenin; (C) binding between AITC and NRF2-NFE2. DG: Gibbs’ free energy; IC50 (t): Theoretical half-inhibitory concentration (calculated using DG).
- Page 10, Lines 323-326: In silico inhibition evaluation of the binding affinity between benzyl isothiocyanate (BITC) and molecular targets of colorectal cancer such. A) Binding between BITC and nuclear factor kB (NF-kB); (B) Binding between BITC and b-catenin; (C) binding between BITC and NRF2-NFE2. DG: Gibbs’ free energy; IC50 (t): Theoretical half-inhibitory concentration (calculated using DG).
- Reviewer: How do you determine the BITC showed stronger affinity from figure 7b compared with figure 7a?? Please explain further about this.
- Authors’ response: It has been proposed that the lower Gibbs’ energy is, the highest affinity. Since we changed the in silico procedures to actual inhibition, this would be an inhibition affinity. We have explained this in the revised manuscript.
Revised manuscript
- Page 8, Lines 306-307: considering that DG values for BITC are lower than AITC.
- Pages 10-11, Lines 328-330: BITC has a stronger affinity for the membrane’s lipids than AITC (based on its lowest DG values), suggesting more interactions between this molecule and the membrane.
- Reviewer: There are two figure 7 in the manuscript. Please revise the last Figure.
- Authors’ response: We apologize with the reviewer for this. We have corrected it in the revised manuscript.
Revised manuscript
- Page 11, Line 349: compared to AITC (Figure 8C).
- Page 12, Line 355: Figure 8.
- Reviewer: Suggest to shift the contents on line 306-312 after figure 7.
- Authors’ response: Corrected as suggested. Please refer to the revised manuscript.
Revised manuscript
- Page 11, Lines 338-344: As shown by the predicted bioavailability values (Figure 8), AITC is more suitable for oral bioavailability as all the evaluated parameters fit the human oral bioavailability ideal conditions highlighted in the colored zone (Figure 8A). In contrast, BITC did not accomplish the ideal unsaturation values. Despite differences between both compounds for potential absorption in the Caco-2 model (Caco-2 M), human intestinal absorption (HIA), and human oral bioavailability (HOB), where AITC displayed the highest values, both compounds received the same bioavailability score (BS: 0.55) (Figure 8B).
- Reviewer: Please double check on the correct figure number being used, revise line 321.
- Authors’ response: Corrected (Page 11, Line 349).

Reviewer 2 Report
The topic of research is interesting, however there is a lack of novelty in the present manuscript.
Although the authors declare throughout the manuscript:
There are no reports of the AITC contents in Raphanus sativus, but black radish (Raphanus sativus var. niger)
Line 367 - There are no reports about using isothiocyanates extracts from cauliflower and radish on colon cancer cell lines.
L 406 No information has been reported on the impact of CIE and RIE on colorectal cancer cell lines.
The above is not totally correct. I suggest performing a review on the topic, several papers are related to isothiocyanates & cancer & cruciferous vegetables.
Parts of the introduction are from web pages that are not properly referenced by the authors.
As an example, authors say: L59-60
During food transformations, chewing, and digestion, glucosinolates break down to form biologically active compounds such as isothiocyanates, indoles, thiocyanates, and nitriles [7]
[7]. Wang, Z.; Kwan, M.L.; Pratt, R.; Roh, J.M.; Kushi, L.H.; Danforth, K.N.; Zhang, Y.; Ambrosone, C.B.; Tang, L. Effects of 504 Cooking Methods on Total Isothiocyanate Yield from Cruciferous Vegetables. Food Sci. Nutr. 2020, 8, 5673–5682, 505 doi:10.1002/fsn3.1836
The authors picked the above from:
https://www.cancer.gov/about-cancer/causes-prevention/risk/diet/cruciferous-vegetables-fact-sheet
During food preparation, chewing, and digestion, the glucosinolates in cruciferous vegetables are broken down to form biologically active compounds such as indoles, nitriles, thiocyanates, and isothiocyanates (1). Hayes JD, Kelleher MO, Eggleston IM. The cancer chemopreventive actions of phytochemicals derived from glucosinolates. European Journal of Nutrition 2008;47 Suppl 2:73-88.
The linearity, matrix effect, the limits of detection (LODs), the limits of quantification (LOQs), recoveries and the relative standard deviation (RSD) of the method to measurement the isothiocyanates in the extracts were not validated.
It is not clear how many samples were analyzed. Only one of each with 3 replicates?
Furthermore only two peaks were identified by chromatographic analysis. And the other ones? Which was the standard used by the authors to check recoveries?
In silico analysis are not necessary at this point. No new information is presented. Check the avalaible literature in the topic.
Figure 2c, the scale need to be correctly presented (0 to 100%).
L468-470
Since benzyl isothiocyanate and allyl isothiocyanate are some of the most abundant isothiocyanates in the evaluated extracts, most of the observed biological activities could be linked to these compounds, aimed by their high bioavailability
The above conclusion is not supported by the results obtained. Bioavailability was researched ¿
The manuscript contains too many references that are not necessary.
Author Response
Authors’ responses to Reviewer 2
- Reviewer: The topic of research is interesting, however there is a lack of novelty in the present manuscript. Although the authors declare throughout the manuscript:
There are no reports of the AITC contents in Raphanus sativus, but black radish (Raphanus sativus var. niger)
Line 367 - There are no reports about using isothiocyanates extracts from cauliflower and radish on colon cancer cell lines.
L 406 No information has been reported on the impact of CIE and RIE on colorectal cancer cell lines.
The above is not totally correct. I suggest performing a review on the topic, several papers are related to isothiocyanates & cancer & cruciferous vegetables.
- Authors’ response: Thank you for your comments. We have made a review on the topic, and we have deleted some of the sentences. Moreover, the introduction has been improved and conclusions have been adjusted. Please refer to the revised manuscript.
- Reports about AITC and BITC contents in cauliflower and radish
The reviewer is correct regarding AITC and BITC contents in radish. We have found some reports regarding the main glucosinolate in Raphanus sativus (e.g., Allyl glucosinolate, n-hexyl-glucosinolate). However, these reports have not considered the quantification of allyl isothiocyanate in these plant species (Fahey et al., 2001). Kjaet et al. (1978) reported the presence of 4-methylthio-3-butenyl isothiocyanate in fresh daikon or white Japanese radish (R. sativus) (Blažević & Mastelić, 2009). Despite being evaluated in daikon, no AITC or BITC contents were found (Marchioni et al., 2021). Beevi et al. (2009) reported that R. sativus extracts contained 0.013-0.21 mg/g AITC and 0.001-0.023 mg/g BITC in root, stem, and leaf, but the root and leaves contained the highest AITC and BITC levels. More recently, Liu et al. (2018) reported a complete characterization of plant volatiles from several Brassicaceae vegetables, including radish and cauliflower, and reported normalized levels of AITC in these vegetables without indicating its exact amount. We have added this later information in the revised manuscript.
- Reports regarding the anti-cancer potential of AITC and BITC on colon cancer cell lines
We extensively reviewed the literature as suggested, and we did not find experiments evaluating R. sativus or B. oleracea var. Botrytis extracts on colorectal cancer cell lines. However, since several authors have reported the effect of glucosinolates or pure standard in colon cancer cell lines, we decided to delete the sentence. Moreover, we are aware that more research is needed to fully indicate a chemopreventive effect from these extracts. Nair et al. (2020) reported the anti-cancer effect of sinigrin and prepared fractions, extracted from R. sativus roots on several cancer cell lines such as prostate cancer (DU-145), colon adenocarcinoma (HCT-15), and melanoma (A-375). For the HCT-15 cells, the authors reported IC50 values of 16.76 µg/mL and 21.42 µg/mL for sinigrin and sinigrin-rich root extract, respectively. Additional experiments evaluating the number of apoptotic cells and caspase-3 activity were evaluated in DU-145 cells, as the extracts showed the lowest IC50 for this cell line. In HT-29 cells, AITC and BITC displayed IC50: 5.4 µM and 10.0 µM, respectively (Zhang et al., 2003).
We have included part of the presented information in the revised manuscript.
Revised manuscript:
- Page 13, Lines 402-406: . The evaluation of root, leaves, and stem acetone and hexane extracts of sativus yielded 0.013-0.21 mg/g AITC and 0.001-0.023 mg/g BITC [45]. More recently, Liu et al. [46] indicated that cauliflower contained some of the highest AITC levels among 18 Brassicaceae plants, whereas radish displayed some of the lowest levels for the same isothiocyanate.
- Page 14, Lines 438-440: On HT-29 cells, AITC and BITC displayed IC50: 5.4 µM (53.5 ´ 10-3 mg/mL) and 10 µM (1.49´ 10-3 mg/mL), respectively [53].
References
Blažević, I., & Mastelić, J. (2009). Glucosinolate degradation products and other bound and free volatiles in the leaves and roots of radish (Raphanus sativus L.). Food Chemistry, 113(1), 96–102. https://doi.org/10.1016/j.foodchem.2008.07.029
Fahey, J. W., Zalcmann, A. T., & Talalay, P. (2001). The chemical diversity and distribution of glucosinolates and isothiocyanates among plants. Phytochemistry, 56(1), 5–51. https://doi.org/10.1016/S0031-9422(00)00316-2
Kjær, A., Øgaard Madsen, J., Maeda, Y., Ozawa, Y., & Uda, Y. (1978). Volatiles in Distillates of Fresh Radish of Japanese and Kenyan Origin. Agricultural and Biological Chemistry, 42(9), 1715–1721. https://doi.org/10.1080/00021369.1978.10863235
Marchioni, I., Martinelli, M., Ascrizzi, R., Gabbrielli, C., Flamini, G., Pistelli, L., & Pistelli, L. (2021). Small Functional Foods: Comparative Phytochemical and Nutritional Analyses of Five Microgreens of the Brassicaceae Family. Foods, 10(2), 427. https://doi.org/10.3390/foods10020427
Nair, A. B., Gandhi, D., Patel, S. S., Morsy, M. A., Gorain, B., Attimarad, M., & Shah, J. N. (2020). Development of HPLC Method for Quantification of Sinigrin from Raphanus sativus Roots and Evaluation of Its Anticancer Potential. Molecules, 25(21), 4947. https://doi.org/10.3390/molecules25214947
Zhang, Y., Tang, L., & Gonzalez, V. (2003). Selected isothiocyanates rapidly induce growth inhibition of cancer cells. Molecular Cancer Therapeutics, 2(10), 1045–1052. https://aacrjournals.org/mct/article/2/10/1045/234075/Selected-isothiocyanates-rapidly-induce-growth
- Reviewer: Parts of the introduction are from web pages that are not properly referenced by the authors.
As an example, authors say: L59-60
During food transformations, chewing, and digestion, glucosinolates break down to form biologically active compounds such as isothiocyanates, indoles, thiocyanates, and nitriles [7]
[7]. Wang, Z.; Kwan, M.L.; Pratt, R.; Roh, J.M.; Kushi, L.H.; Danforth, K.N.; Zhang, Y.; Ambrosone, C.B.; Tang, L. Effects of 504 Cooking Methods on Total Isothiocyanate Yield from Cruciferous Vegetables. Food Sci. Nutr. 2020, 8, 5673–5682, 505 doi:10.1002/fsn3.1836
The authors picked the above from:
https://www.cancer.gov/about-cancer/causes-prevention/risk/diet/cruciferous-vegetables-fact-sheet
During food preparation, chewing, and digestion, the glucosinolates in cruciferous vegetables are broken down to form biologically active compounds such as indoles, nitriles, thiocyanates, and isothiocyanates (1). Hayes JD, Kelleher MO, Eggleston IM. The cancer chemopreventive actions of phytochemicals derived from glucosinolates. European Journal of Nutrition 2008;47 Suppl 2:73-88.
- Authors’ response: We apologize with the reviewer, and we have added the correct references for these sentences. We have also improved the introduction. Please refer to the revised manuscript.
Revised manuscript
- Page 2, Lines 49-52: Among cruciferous vegetables, the Brassicaceae family is one of the most notorious because it groups some of the most consumed vegetables globally and provide significant health benefits [5]. Some of the most notorious Brassicaceae are cauliflower (Brassica oleracea botrytis), broccoli (B. oleracea var. italica),
- Page 2, Lines 62-63: During the physiological transformation of glucosinolates-rich foods, biologically active compounds such as isothiocyanates, among other components, are formed [6].
- Page 2, Lines 72-77: Particularly for radish, 4-methylthio-3-butenyl isothiocyanate is considered the main ITC, but other compounds such as AITC, BITC, and phenetyl isothiocyanate have also been linked to its pungent flavor and biological properties [11]. Regarding cauliflower, its odor and the typical flavor are associated to volatile compounds, ITCs among them, such as AITC, 2-methylpropyl isothiocyanate, but-3-enyl isothiocyanate, 3-methylbutyl isothiocyanate, and 4-methylpentyl isothiocyanate [12].
References
Hayes, J. D., Kelleher, M. O., & Eggleston, I. M. (2008). The cancer chemopreventive actions of phytochemicals derived from glucosinolates. European Journal of Nutrition, 47(S2), 73–88. https://doi.org/10.1007/s00394-008-2009-8
Liu, Y., Zhang, H., Umashankar, S., Liang, X., Lee, H., Swarup, S., & Ong, C. (2018). Characterization of Plant Volatiles Reveals Distinct Metabolic Profiles and Pathways among 12 Brassicaceae Vegetables. Metabolites, 8(4), 94. https://doi.org/10.3390/metabo8040094
Suh, S.-J., Moon, S.-K., & Kim, C.-H. (2006). Raphanus sativus and its isothiocyanates inhibit vascular smooth muscle cells proliferation and induce G1 cell cycle arrest. International Immunopharmacology, 6(5), 854–861. https://doi.org/10.1016/j.intimp.2005.11.014
Valette, L., Fernandez, X., Poulain, S., Lizzani-Cuvelier, L., & Loiseau, A.-M. (2006). Chemical composition of the volatile extracts fromBrassica oleracea L. var.botrytis ‘Romanesco’ cauliflower seeds. Flavour and Fragrance Journal, 21(1), 107–110. https://doi.org/10.1002/ffj.1530
- Reviewer: The linearity, matrix effect, the limits of detection (LODs), the limits of quantification (LOQs), recoveries and the relative standard deviation (RSD) of the method to measurement the isothiocyanatesin the extracts were not validated.
Authors’ response: Thanks. We have added this information in the revised manuscript as a supplementary information. Please refer to the revised manuscript.
Revised manuscript
- Page 3, Lines 138-139: The validation of the conducted HPLC method is shown in Supplementary Table S1 and Supplementary Table S2.
Supplementary information
Supplementary Table 1. Validation of HPLC method using for the identification and quantification of isothiocyanates.
|
Standard name |
Detection wavelength (nm) |
RT (min) |
Linearity range (mg/mL) |
Regression coefficient (R2) |
LOD (mg/mL) |
LOQ (mg/mL) |
|
AITC |
240 |
21.61 |
0-100 |
0.991 |
2.05 |
2.24 |
|
BITC |
240 |
39.02 |
0-100 |
0.996 |
2.10 |
2.38 |
AITC: Allyl isothiocyanate; BITC: benzyl isothiocyanate; RT: retention time; LOD: Limit of detection; LOQ: limit of quantification.
Supplementary Table 2. Results from the recovery test of isothiocyanates.
|
Standard name |
Amount tested |
Initial amount (mg/ml) |
Found amount (mg/mL) |
Recovery (%) |
Average recovery (%) |
RSD (%) |
|
AITC |
Low |
15.20 |
14.89 |
97.96 |
98.69 |
0.79 |
|
Medium |
35.71 |
35.21 |
98.59 |
|||
|
High |
60.22 |
59.93 |
99.52 |
|||
|
BITC |
Low |
14.90 |
14.88 |
99.87 |
99.27 |
0.64 |
|
Medium |
40.23 |
39.67 |
98.61 |
|||
|
High |
62.14 |
61.72 |
99.32 |
AITC: Allyl isothiocyanate; BITC: benzyl isothiocyanate; RSD: Relative standard deviation.
- Reviewer: It is not clear how many samples were analyzed. Only one of each with 3 replicates? Furthermore, only two peaks were identified by chromatographic analysis. And the other ones? Which was the standard used by the authors to check recoveries?
Authors’ response: Thanks. Three extracts of each plant were obtained an analyzed, and three replicates were obtained from each one. We focused on AITC and BITC as our target isothiocyanates, and we did not assay other standards than the ones presented in this research, considering that AITC and BITC showed the most prominent peaks. To check recoveries, we used AITC and BITC, and presented the revised Supplementary Information file.
Revised manuscript
- Page 3, Lines 137-139: Three different RIE and CIE extracts were prepared, and replicates of each extract, using two injections, were quantified (n=18). The validation of the conducted HPLC method is shown in Supplementary Table S1 and Supplementary Table S2.
- Reviewer: In silico analysis are not necessary at this point. No new information is presented. Check the available literature in the topic.
Authors’ response: We have conducted a new in silico inhibition analysis, docking each isothiocyanate with the position of target inhibitors for NF-kB, b-catenin, and NRF2-NFE2. Although there is available literature in the topic, we believe the targeted inhibition of these molecular targets using AITC and BITC is novel. Please refer to the revised manuscript in Section 2.5, and revised Figure 6 and Figure 7.
The modulation of the selected cancer targets by AITC and BITC has been reported in several cell lines, but no in silico examinations have been conducted. We found an in silico interaction between AITC and NRF2 protein (Rajakumar et al., 2018a) or NF-kB (Rajakumar et al., 2018b), but the authors docked the structures finding the highest possibility of interaction instead of specifically evaluating the inhibition, which is our new perspective from the in silico studies. In addition, an increased protein expression of NRF2 in AITC-treated HepG2 human hepatoma cells (25 µM) (Jeong et al., 2005). The modulatory effects of AITC in the NF-kB pathway in HT-29 cells were reported by Jeong et. al (2004).
Regarding the available literature, it has been proposed that BITC has several molecular targets such as macrophage migration inhibitory factor (MIF), the transient receptor potential cation channel subfamily A member 1 (TRPA1), phosphodiesterase 5A (PDE5A), the protooncogene tyrosine-protein kinase ROS (ROS1), the Muscarinic acetylcholine receptor M1 (CHRM1), and the prostanoid EP4 receptor (EP4R). Similarly, AITC also has molecular targets such as MIF, TRPA1, ROS1, EP4R, and acts as an apoptosis agonist (Guerrero-Alonso et al., 2021).
An overall calculation of the proportion of glucosinolates (%) meeting empirical drug-like criteria such as molecular weight (80-100 %), the partition coefficient octanol/water (clogP) (60-80%), hydrogen-bond acceptors (100 %), hydrogen-bond donors (100 %), the number of heavy atoms (40 %), number of rotatable atoms (70-80 %), polar surface area (PSA) oral (<20 %), PSA blood-brain barrier (40-50 %), and molar refractivity (90-100 %) was presented by Guerrero-Alonso et al. (2021). However, no focus on AITC or BITC was given. Results presented in Figure 8 aim to valorize further bioaccessibility and bioavailability assays involving identifying and quantifying these isothiocyanates along the digestion. We have also included a sentence indicating this purpose. Please refer to the revised manuscript.
Revised manuscript
- Pages 14-15, Lines 469-475: Experimentally, it has been found that AITC inhibited the lipopolysaccharide (LPS) NF-kB activation and induced IkBa phosphorylation in HT-29 cells [65]. The in silico interaction between AITC and NRF2 was explored by Rajakumar et al. [66], and the authors suggested an NRF2 induction by AITC in 7,12-dimethylbenz(a) anthracene (DMBA)-induced mammary tumors in vivo. Similarly, an interaction between AITC and NF-kB was also reported [67], but the authors explained no inhibition perspective.
- Page 15, Lines 493-496: Further research evaluating the bioaccessibility and bioavailability of the targeted isothiocyanates explored in this research is encouraged, aiming to evaluate their natural presence under gastrointestinal conditions, potential release, and biological effects.
References
Guerrero‐Alonso, A., Antunez‐Mojica, M., & Medina‐Franco, J. L. (2021). Chemoinformatic Analysis of Isothiocyanates: Their Impact in Nature and Medicine. Molecular Informatics, 40(11), 2100172. https://doi.org/10.1002/minf.202100172
Jeong, W.-S., Keum, Y.-S., Chen, C., Jain, M. R., Shen, G., Kim, J.-H., Li, W., & Kong, A.-N. T. (2005). Differential Expression and Stability of Endogenous Nuclear Factor E2-related Factor 2 (Nrf2) by Natural Chemopreventive Compounds in HepG2 Human Hepatoma Cells. BMB Reports, 38(2), 167–176. https://doi.org/10.5483/BMBRep.2005.38.2.167
Jeong, W.-S., Kim, I.-W., Hu, R., & Kong, A.-N. T. (2004). Modulatory Properties of Various Natural Chemopreventive Agents on the Activation of NF-κB Signaling Pathway. Pharmaceutical Research, 21(4), 661–670. https://doi.org/10.1023/B:PHAM.0000022413.43212.cf
Rajakumar, T., Pugalendhi, P., Jayaganesh, R., Ananthakrishnan, D., & Gunasekaran, K. (2018). Effect of allyl isothiocyanate on NF-κB signaling in 7,12-dimethylbenz(a)anthracene and N-methyl-N-nitrosourea-induced mammary carcinogenesis. Breast Cancer, 25(1), 50–59. https://doi.org/10.1007/s12282-017-0783-y
Rajakumar, T., Pugalendhi, P., Thilagavathi, S., Ananthakrishnan, D., & Gunasekaran, K. (2018). Allyl isothiocyanate, a potent chemopreventive agent targets AhR/Nrf2 signaling pathway in chemically induced mammary carcinogenesis. Molecular and Cellular Biochemistry, 437(1–2), 1–12. https://doi.org/10.1007/s11010-017-3091-0
- Reviewer: Figure 2c, the scale needs to be correctly presented (0 to 100%).
Authors’ response: We apologize with the reviewer. We have provided a new Figure 2 showing all scales 0-100 %. Please refer to the revised manuscript.
- Reviewer: L468-470: Since benzyl isothiocyanate and allyl isothiocyanate are some of the most abundant isothiocyanates in the evaluated extracts, most of the observed biological activities could be linked to these compounds, aimed by their high bioavailability
The above conclusion is not supported by the results obtained. Bioavailability was researched?
Authors’ response: We appreciate the reviewer’s comments, and we agree with the reviewer. We are currently investigating the bioaccessibility, an initial step to further assess bioavailability of BITC and AITC using simulated gastrointestinal digestion of cauliflower and radish. Our preliminary results have shown a high bioaccessibility of these compounds (35-60 %) in the stomach and small intestine. However, we require additional experiments to assess intestinal permeability and bioavailability using an animal model. We have adjusted the information presented in the indicated lines. Please refer to the revised manuscript.
Revised manuscript:
- Page 15, Lines 515-518: Although it has been reported a high bioavailability of AITC and BITC and biological activities coming from CIE and RIE extracts could be linked to these compounds, further in vivo experiments must be conducted to prove their properties.
- Reviewer: The manuscript contains too many references that are not necessary.
Authors’ response: We apologize with the reviewer. We have deleted the following references:
Angladon, M.A.; Fossépré, M.; Leherte, L.; Vercauteren, D.P. Interaction of POPC, DPPC, and POPE with the μ Opioid Receptor: A Coarse-Grained Molecular Dynamics Study. PLoS One 2019, 14, 1–19, doi:10.1371/journal.pone.0213646.
Arango-Varela, S.S.; Luzardo-Ocampo, I.; Maldonado-Celis, M.E. Andean Berry (Vaccinium Meridionale Swartz) Juice, in Combination with Aspirin, Displayed Antiproliferative and pro-Apoptotic Mechanisms in Vitro While Exhibiting Protective Effects against AOM-Induced Colorectal Cancer in Vivo. Food Res. Int. 2022, 157, 111244, doi:10.1016/j.foodres.2022.111244.
Arango‐Varela, S.S.; Luzardo‐Ocampo, I.; Reyes‐Dieck, C.; Yahia, E.M.; Maldonado‐Celis, M.E. Antiproliferative Potential of Andean Berry (Vaccinium Meridionale Swartz) Juice in Combination with Aspirin in Human SW480 Colon Adenocarcinoma Cells. J. Food Biochem. 2021, 45, doi:10.1111/jfbc.13760.
Caicedo-Lopez, L.H.L.H.; Luzardo-Ocampo, I.; Cuellar-Nuñez, M.L.L.; Campos-Vega, R.; Mendoza, S.; Loarca-Piña, G. Effect of the in Vitro Gastrointestinal Digestion on Free-Phenolic Compounds and Mono/Oligosaccharides from Moringa Oleifera Leaves: Bioaccessibility, Intestinal Permeability and Antioxidant Capacity. Food Res. Int. 2019, 120, 631–642, doi:10.1016/j.foodres.2018.11.017.
Godínez-Oviedo, A.; Cuellar-Núñez, M.L.; Luzardo-Ocampo, I.; Campos-Vega, R.; Hernández-Iturriaga, M. A Dynamic and Integrated in Vitro/Ex Vivo Gastrointestinal Model for the Evaluation of the Probability and Severity of Infection in Humans by Salmonella Spp. Vehiculated in Different Matrices. Food Microbiol. 2021, 95, 103671, doi:10.1016/j.fm.2020.103671.
Herrera-Cazares, L.A.; Luzardo-Ocampo, I.; Ramírez-Jiménez, A.K.; Gutiérrez-Uribe, J.A..; Campos-Vega, R.; Gaytán-Martínez, M. Influence of Extrusion Process on the Release of Phenolic Compounds from Mango (Mangifera Indica L.) Bagasse-Added Confections and Evaluation of Their Bioaccessibility, Intestinal Permeability, and Antioxidant Capacity. Food Res. Int. 2021, 148, 110591, doi:10.1016/j.foodres.2021.110591.

Reviewer 3 Report
In this manuscript the authors reported the investigation of The CIE samples displayed the highest allyl isothiocyanate (AITC) contents, whereas RIE was the most abundant in benzyl isothiocyanate (BITC). Some interesting results are obtained. I therefore recommend an acceptance for publishing after next revisions.
1.Pages 2, abstract part, some background sentences can be simplified;
2.Introduction part, if possible, some important and relative reports should be added to show clear background;
3. Some minor Language error and style should be modified;
Author Response
Authors’ responses to Reviewer 3
- Reviewer: In this manuscript the authors reported the investigation of The CIE samples displayed the highest allyl isothiocyanate (AITC) contents, whereas RIE was the most abundant in benzyl isothiocyanate (BITC). Some interesting results are obtained. I therefore recommend an acceptance for publishing after next revisions.
Authors’ response: We appreciate the reviewer’s comments.
- Reviewer: Pages 2, abstract part, some background sentences can be simplified.
Authors’ response: Thanks. Sentences have been simplified in the revised abstract.
Revised manuscript
- Page 1, Lines 17-18: Cruciferous vegetables such as cauliflower and radish contain isothiocyanates exhibiting chemoprotective effects in vitro and in vivo. This research aimed to assess
- Reviewer: Introduction part, if possible, some important and relative reports should be added to show clear background.
Authors’ response: Thanks. Introduction has been improved. Please refer to the revised manuscript.
Revised manuscript
- Page 2, Lines 49-52: Among cruciferous vegetables, the Brassicaceae family is one of the most notorious because it groups some of the most consumed vegetables globally and provide significant health benefits [5]. Some of the most notorious Brassicaceae are cauliflower (Brassica oleracea botrytis), broccoli (B. oleracea var. italica),
- Page 2, Lines 62-63: During the physiological transformation of glucosinolates-rich foods, biologically active compounds such as isothiocyanates, among other components, are formed [6].
- Page 2, Lines 72-77: Particularly for radish, 4-methylthio-3-butenyl isothiocyanate is considered the main ITC, but other compounds such as AITC, BITC, and phenetyl isothiocyanate have also been linked to its pungent flavor and biological properties [11]. Regarding cauliflower, its odor and the typical flavor are associated to volatile compounds, ITCs among them, such as AITC, 2-methylpropyl isothiocyanate, but-3-enyl isothiocyanate, 3-methylbutyl isothiocyanate, and 4-methylpentyl isothiocyanate [12].
References
Hayes, J. D., Kelleher, M. O., & Eggleston, I. M. (2008). The cancer chemopreventive actions of phytochemicals derived from glucosinolates. European Journal of Nutrition, 47(S2), 73–88. https://doi.org/10.1007/s00394-008-2009-8
Liu, Y., Zhang, H., Umashankar, S., Liang, X., Lee, H., Swarup, S., & Ong, C. (2018). Characterization of Plant Volatiles Reveals Distinct Metabolic Profiles and Pathways among 12 Brassicaceae Vegetables. Metabolites, 8(4), 94. https://doi.org/10.3390/metabo8040094
Suh, S.-J., Moon, S.-K., & Kim, C.-H. (2006). Raphanus sativus and its isothiocyanates inhibit vascular smooth muscle cells proliferation and induce G1 cell cycle arrest. International Immunopharmacology, 6(5), 854–861. https://doi.org/10.1016/j.intimp.2005.11.014
Valette, L., Fernandez, X., Poulain, S., Lizzani-Cuvelier, L., & Loiseau, A.-M. (2006). Chemical composition of the volatile extracts fromBrassica oleracea L. var.botrytis ‘Romanesco’ cauliflower seeds. Flavour and Fragrance Journal, 21(1), 107–110. https://doi.org/10.1002/ffj.1530
- Reviewer: Some minor Language error and style should be modified.
Authors’ response: Thanks. We have double checked the manuscript to improve language error and style. Please refer to the revised manuscript.
Revised manuscript
- Page 1, Lines 42-43: synthesized anticancer drugs has been shown.
- Page 2, Line 44: The search for an.
- Page 2, Line 58: phenolics acids; carotenoids such as b-carotene.
- Page 2, Line 61: which are responsible for the.
- Page 2, Lines 66-67: Dietary isothiocyanates (ITCs) are biochemical compounds that can modify certain pathways.
- Page 2, Line 82: which is involved in inflammation processes.
- Page 2, Line 85: such as histone deacetylation and
- Page 2, Line 90: induced apoptotic cell death by inhibiting
- Page 2, Line 95: or radish isothiocyanates
- Page 3, Lines 105-107: for seven Subsequently, the freeze-dried material was ground and stored in sealed bags at -80 ºC until further analyses were carried out.
- Page 4, Line 180: 680 nm was used to.
- Page 5, Line 208: was modeled
- Page 5, Line 232: compounds were
- Page 5, Line 244: isothiocyanates
- Page 6, Lines 267-268: AITC being the most effective in HCT116 and BITC on HT-29. Oxaliplatin (OX) was evaluated as representative
- Page 8, Line 289: for each cell line between.
- Page 8, Lines 311-312: However, a higher BITC amount is required to induce a response with NF-kB (Figure 6) potentially.
- Page 11, Line 350: diagram for predicting the
- Page 13, Line 371: (thioglucosidase glycohydrolase) catalyzes
- Page 13, Line 375: of glucosinolates and the main
- Page 13, Line 377: isothiocyanates, exhibiting many
- Page 13, Lines 387-388: Human exposure to AITC is widespread and abundant and its biological properties, such as anti-cancer effects,
- Page 13, Lines 398-401: There have been reported AITC contents in Raphanus sativus niger (0.0025 % of AITC) and butyl isothiocyanate-containing essential oil. In another report, black radish was reported to have 14.5-23.1 µg AITC/mL for each 2-8 g powder/100 mL (not specified).
- Page 13, Lines 409-411: However, it is one of the predominant isothiocyanates in cauliflower and radish (4.5 and 2.2 %, respectively) of the total myrosinase-mediated glucosinolate hydrolysis products [50].
- Page 13, Line 413: and radish (RIE) isothiocyanates
- Page 14, Line 422: stage, as
- Page 14, Lines 454-460: Hence, both extracts and the standards could induce ROS, although CIE and BITC effects were more substantial on HCT116, which could be related to major BITC contents in CIE. There are no reports on the ROS-inducing effects of CIE and RIE or isothiocyanates extracts on colorectal cancer cell lines. Nevertheless, the challenge of human metastatic bone PC3 cells with cauliflower seed extracts showed an increased ROS effect [61]. Sulforaphane from broccoli, a class of glucosinolates, can trigger apoptosis in several cancer cell lines and increased ROS levels to induce
- Page 14, Line 462: was partially responsible
- Page 15, Line 504: gene regulation, and
- Page 15, Line 512: The results obtained from this research.

Reviewer 4 Report
Dear colleagues,
The article is very interesting, but it could be improved by clear detection of selective influence for colorectal cells (according to title) as tumoral factors which are described could be presented in other tumors also (as there is no mention in conclusion about it).
Secondly, the abstract and conclusion must be supported by digital data.
Author Response
Authors’ responses to Reviewer 4
- Reviewer: Dear colleagues, the article is very interesting, but it could be improved by clear detection of selective influence for colorectal cells (according to title) as tumoral factors which are described could be presented in other tumors also (as there is no mention in conclusion about it).
Authors’ response: Thank you for your comments. The reviewer is correct since the evaluated assays are not exclusive for colorectal cancer cells. We have added this information in the revised conclusions.
Revised manuscript
- Page 1, Lines 4-5 (title): and Induced ROS in Selected Human HCT116 and HT-29 Colorectal Cancer Cells.
- Page 1, Line 20: and LDH production of selected human colorectal adenocarcinoma cells.
- Page 15, Lines 521-522: Moreover, since the evaluated assays are not exclusive for colorectal cancer cells, additional research involving other cancer cell types are needed.
- Reviewer: Secondly, the abstract and conclusion must be supported by digital data.
Authors’ response: Thank you for your comments. We have improved the abstract with data supporting some of the statements, and the conclusions. Please refer to the revised manuscript.
Revised manuscript:
- Page 1, Lines 17-31: Cruciferous vegetables such as cauliflower and radish contain isothiocyanates exhibiting chemoprotective effects in vitro and in vivo. This research aimed to assess the impact of cauliflower (CIE) and radish (RIE) isothiocyanates’ extracts on the metabolic activity, intracellular reactive oxygen species (ROS), and LDH production of selected human colorectal adenocarcinoma cells (HCT116 and HT-29 for early and late colon cancer development, respectively). Non-cancerous colon cells (CCD-33Co) were used as cytotoxicity control. The CIE samples displayed the highest allyl isothiocyanate (AITC: 12.55 mg/g) contents, whereas RIE was the most abundant in benzyl isothiocyanate (BITC: 15.35 mg/g). Both extracts effectively inhibited HCT116 and HT-29 metabolic activity, but CIE impact was higher than RIE on HCT116 (IC50: 0.56 mg/mL). Assays using the half-inhibitory concentrations (IC50) of all treatments, including AITC and BITC, displayed increased (p<0.05) LDH (absorbance: 0.25-0.40 nm) and ROS release (1190-1697 relative fluorescence units) in both cell lines. BITC showed the highest in silico binding affinity with all the tested colorectal cancer molecular markers (NF-kB, b-catenin, and NRF2-NFE2). Theoretical evaluation of AITC and BITC bioavailability showed high values for both compounds. Results indicated that CIE and RIE extracts display chemopreventive effects in vitro, but additional experiments are needed to validate their effects.
- Page 15, Lines 512-522: The results obtained from this research suggested that cauliflower and radish isothiocyanates’ extracts effectively inhibited HT-29 and HCT116 metabolic activity (>80 %), increased the production of intracellular ROS (41.63-93.29 %, compared to untreated cells)), and induced LDH release (+21.02 to 77.23 %, compared to untreated cells). Although it has been reported a high bioavailability of AITC and BITC and biological activities coming from CIE and RIE extracts could be linked to these compounds, further in vivo experiments must be conducted to prove their properties. The in silico modeling of individual isothiocyanates with molecular cancer targets could suggest a potential inhibition of NF-kB, b-catenin, and NRF2 proteins, but additional in vitro and in vivo experiments are needed to validate these effects. Moreover, since the evaluated assays are not exclusive to colorectal cancer cells, additional research involving other cancer cell types is needed.

Round 2
Reviewer 2 Report
The authors must specify and explain what methodology they have followed for the validation of the analytical method. Thus, although the authors present information required in my first review regarding the limits of detection, quantification and recovery; they do not explain how it was done (methodology or guide followed). Beyond that, if the detection limit of the analytical method is established at 2.05 (mg/mL) for the AITC and 2.10 (mg/mL) for the BITC; considering the low point for the recovery studies of 15.20 and 14.90 is not appropriate. The authors use the AITC and the BITC as standards. When internal standardization is used in recovery experiments the surrogate should be an entity chemically distinct from the analytes, and therefore will not have identical chemical properties. It is unclear what procedure they used for the extraction studies of the analytes of interest from the matrix.
Author Response
- Reviewer: The authors must specify and explain what methodology they have followed for the validation of the analytical method. Thus, although the authors present information required in my first review regarding the limits of detection, quantification, and recovery; they do not explain how it was done (methodology or guide followed).
- Author's Response: Thank you for your comments. We have specified the conducted methodology in the revised manuscript. The procedure was conducted accordingly to the International Council for Harmonisation of Technical Requirements for Pharmaceuticals for Human Use (ICH) and reported HPLC validation procedures of these two standards. Please refer to the revised manuscript.
Revised manuscript:
Pages 3-4, Lines 143-162: Validation was carried out as recommended by the International Council for Harmonization (ICH) [22]. Briefly, linearity was assessed using a working range of each standard (0-100 µg/mL), and each concentration was injected three times to obtain the area under the curve (AUC). Data was used to plot a linear regression curve (y=mx+b, where “y” is the response or AUC, “x” is the assayed concentration of AITC or BITC; “m” is the slope of the curve, and “b” is the y-intercept). The limit of detection (LOD) and limit of quantification (LOQ) were considered the limits at which AITC or BITC can be reliably detected or quantified using a 3:1 signal-to-noise ratio, considering the standard deviation of the calibration curve (s) and the slope of the calibration curve (S), LOD=3.3s/S and LOQ=10s/S. For determining the recovery, a minimum of 6 determination of the 100% of the test concentrations (low concentration: 15.20 and 14.90 µg/mL; medium concentration: 35.71 and 40.23 µg/mL; and high concentration: 60.22 and 62.14 µg/mL) were prepared, spiked into the processed sample as indicated in 2.2 section, and injected in the HPLC system. The AUC for each test was used to calculate the AITC or BITC concentrations using the previously calculated standard curves. The percentual recovery and the relative standard deviation (%RSD) were also calculated. For each assay, separate weights of the standards were evaluated. Together with the suggested procedures of ICH, the usage of AITC and BITC standards was based on previous reports using these pure standards spiked into the sample for recovery assays [23,24].
References
Patel, M., Raval, M., Joshi, M., & Sanandia, J. (2018). High-performance thin-layer chromatography and reversed-phase high-performance liquid chromatography methods for fingerprinting of Salvadora persica root powder extract using benzyl isothiocyanate as biomarker. JPC - Journal of Planar Chromatography - Modern TLC, 31(6), 445–450. https://doi.org/10.1556/1006.2018.31.6.4
Agrawal, S., Yallatikar, T., & Gurjar, P. (2019). Reversed-phase high-performance liquid chromatographic method development and validation for allyl isothiocyanate estimation in phytosomes of Brassica nigra extract. Journal of Advanced Pharmaceutical Technology & Research, 10(3), 126. https://doi.org/10.4103/japtr.JAPTR_382_18
ICH. (2022). Validation of Analytical Procedures Q2 (R2). International Council for Harmonisation of Technical Requirements for Pharmaceuticals for Human Use. https://www.ema.europa.eu/en/documents/scientific-guideline/ich-guideline-q2r2-validation-analytical-procedures-step-2b_en.pdf
- Reviewer: Beyond that, if the detection limit of the analytical method is established at 2.05 (mg/mL) for the AITC and 2.10 (mg/mL) for the BITC; considering the low point for the recovery studies of 15.20 and 14.90 is not appropriate.
Authors’ response: We apologize with the reviewer since when we copied and pasted our answer, previously elaborated in a Word document, concentrations in micrograms/mL (µg/mL) were pasted as milligrams/mL. Based on this, our lowest concentrations for both AITC and BITC are higher than LOD of both compounds.
- Reviewer: The authors use the AITC and the BITC as standards. When internal standardization is used in recovery experiments the surrogate should be an entity chemically distinct from the analytes, and therefore will not have identical chemical properties.
- Authors’ response: We appreciate the reviewer's comment. For instance, we have observed that HPLC validation procedures for certain chemical substances require the use of surrogates (e.g., caffeine, using theobromine, paraxanthine, and theophylline). However, for AITC and BITC authors have used exactly these compounds for recovery experiments as reported by Agrawal et al. (2019) and Patel et al. (2018) for AITC and BITC, respectively. Hence, we believe using AITC and BITC for HPLC recovery is correct.
- Reviewer: It is unclear what procedure they used for the extraction studies of the analytes of interest from the matrix.
- Author's response: We apologize to the reviewer if the described procedure was incorrect. We have added more information to make it clearer. Please refer to the revised manuscript.
Revised manuscript:
Page 3, Lines 108-127:
2.2. Glucosinolates extraction and enzymatic conversion into isothiocyanates
The extraction of glucosinolates was conducted, as reported by Förster et al. [20]. Briefly, the freeze-dried material (20 mg cauliflower heads or radish roots) was mixed with 750 µL of 70 % v/v methanol, and the mixture was placed in 1.5 mL microcentrifuge tubes with a locking snap cap and heated at 80 ºC for 10 min in a Thermomixer (Eppendorf, Hamburg, Germany). The extraction was repeated twice using 500 µL each time.
After cooling (22 ± 1 ºC), the samples were centrifuged (16000´ g), and supernatants were concentrated in a vacuum concentrator (Savant SpeedVac DNA130, ThermoScientific, Waltham, MA, USA) up to a 150 µL volume. Then, 200 µL of 0.4 M barium acetate was added, and the volume was completed up to 1 mL using MilliQ water. The samples were incubated for 30 min at room temperature (22 ± 1 ºC) and filtered through centrifugation using 0.22 µm Costar Spin X tubes (CLS8160, Sigma-Aldrich, Sant Louis, MO, US) in a centrifuge (16000´ g). Then, one milliliter of the filtrate was incubated with 0.05 U myrosinase (thioglucosidase from Sinapis alba, Sigma-Aldrich) for 8 h at 37 ºC to allow enzymatic conversion from the original glucosinolates (sinigrin and glucotropaeolin) to their derived isothiocyanates (AITC and BITC, respectively). The resulting hydrolyzed samples (glucosinolates extracts) were filtered again by centrifugation (16000´ g) using 0.22 µm Costar Spin X tubes, and the filtrates were stored at -80 ºC to further analysis. Extracts from cauliflower and radish were named cauliflower isothiocyanate (CIE) and radish isothiocyanate (RIE) extracts, respectively.
